# Oral Candidosis: Pathophysiology and Best Practice for Diagnosis, Classification, and Successful Management

**DOI:** 10.3390/jof7070555

**Published:** 2021-07-13

**Authors:** Shin-Yu Lu

**Affiliations:** Oral Pathology and Family Dentistry Section, Department of Dentistry, Kaohsiung Chang Gung Memorial Hospital and Chang Gung University College of Medicine, 123 Dapi Road, Niaosong District, Kaohsiung 833, Taiwan; jasminelu@adm.cgmh.org.tw

**Keywords:** oral candidosis, systemic candidosis, hematinic deficiency, anemia, immunosuppression, cell-mediated immunity

## Abstract

Oral candidosis is the most common fungal infection that frequently occurs in patients debilitated by other diseases or conditions. No candidosis happens without a cause; hence oral candidosis has been branded as a disease of the diseased. Prior research has identified oral candidosis as a mark of systemic diseases, such as hematinic deficiency, diabetes mellitus, leukopenia, HIV/AIDS, malignancies, and carbohydrate-rich diet, drugs, or immunosuppressive conditions. An array of interaction between *Candida* and the host is dynamic and complex. *Candida* exhibits multifaceted strategies for growth, proliferation, evasion of host defenses, and survival within the host to induce fungal infection. Oral candidosis presents a variety of clinical forms, including pseudomembranous candidosis, erythematous candidosis, angular cheilitis, median rhomboid glossitis, cheilocandidosis, juxtavermillion candidosis, mucocutaneous candidosis, hyperplastic candidosis, oropharyngeal candidosis, and rare suppurative candidosis. The prognosis is usually favorable, but treatment failure or recurrence is common due to either incorrect diagnosis, missing other pathology, inability to address underlying risk factors, or inaccurate prescription of antifungal agents. In immunocompromised patients, oropharyngeal candidosis can spread to the bloodstream or upper gastrointestinal tract, leading to potentially lethal systemic candidosis. This review therefore describes oral candidosis with regard to its pathophysiology and best practice for diagnosis, practical classification, and successful management.

## 1. Introduction

*Candida* species can be both commensals and opportunistic pathogens of the oral cavity [1]. The genus *Candida* includes more than 150 species of asporogenous yeast that are widespread in the environment [2]. Most species cannot live at the temperature of the human body [2,3], so only a limited number of *Candida* species harmlessly live in the healthy mouth, eyes, gastrointestinal tract, vagina, and skin [2,3,4,5,6,7,8,9]. The prevalence of *Candida albicans* (*C. albicans*) isolated from the oral cavity are reported to be around 30% to 50% of the healthy general population, 50–65% of people who wear dentures, 65–88% of people who reside in long term care facilities, and 90–95% of individuals with human immunodeficiency virus (HIV) infection and also in patients who receive corticosteroid, chemotherapy, immunosuppressant, or radiation for head and neck cancers [1,2,4,10,11,12,13]. When an imbalance in the normal oral flora or inadequate immune defense occurs, *Candida* can overgrow to induce a fungal infection called oral candidosis or thrush. The terms oral candidosis and candidiasis are synonymous. In this study, the term candidosis is used in preference to candidiasis because the suffix “-osis” is consistent with the ending used for the vast majority of fungal infections, whereas the ending “-iasis” is used for parasitic diseases [5].

Over 17 *Candida* species can cause oral mucosa and deep tissue infection. Although *C. albicans* remains the most frequent pathogen of candidosis, increased incidence of isolation of non-albicans *Candida* (NAC) species from oral lesions and the bloodstream has subsequently developed [4,5,6,7,14,15,16,17]. The most prominent have been *C. glabrata* and *C. parapsilosis,* followed by *C. tropicalis*
*and C. Krusei*. The four *Candida* species plus *C. albicans* account for more than 90% of cases of oral or disseminated candidosis and candidemia [5,6,7,8,9,10]. This shift in species with a higher incidence of candidosis caused by NAC species has emerged as a major bearing on the morbidity and management of patients who are cared for in intensive care units (ICUs) or have important risk factors, such as hematologic malignancies, transplants, major abdominal surgery, and/or prolonged treatment with corticosteroid, as the increasing antifungal resistance along with a wide range of minimum inhibitory concentrations (MICs) to fluconazole between *Candida* species exists [16,17,18,19,20,21]. All *C.*
*krusei* are intrinsically fluconazole resistant, and infections caused by this species are strongly associated with prior fluconazole prophylaxis and neutropenia [15,16,17,18,19,20,21]. An increasing proportion of *C*. *glabrata* is fluconazole resistant. *C. tropicalis* and *C. parapsilosis* are generally susceptible to azoles; however, *C. tropicalis* is less susceptible to fluconazole than is *C. albicans*. In addition, species distribution and echinocandin resistance show considerable differences between medical centers and countries [16,17,21]. Patient characteristics and prior antifungal therapy also have a considerable influence on the distribution and frequency of *Candida* species regardless of geographical area.

Oral candidosis has received considerable attention since the advent of HIV infection and the increasing prevalence of compromised individuals due to modern therapeutic advances. Oral candidosis is actually a mixed *Candida* species infection. The three major species of *C. albicans* (the commonest; over 50%), *C. tropicalis,* and *C. glabrata* account for more than 80% of oral candidosis because they can evade antimicrobial activity of lysozyme of saliva and easily adhere to denture acrylic surfaces [5,6,7,22]. Tobgi and Samaranayake and MacFarlane reported a significant dose-response relationship between lysozyme concentration and fungicidal activity. When different species of *Candida* were tested, *C. krusei* and *C. parapsilosis* were found to be most sensitive to lysozyme compared with *C. albicans* and *C. glabrata,* which were least sensitive [22]. The relatively high resistance of *C. albicans* and *C. glabrata* to lysozyme may partly explain their high oral prevalence both among carriers and in patients with oral candidosis [22,23,24]. The mixed colonization of *C. albicans* and *C. glabrata* can enhance fungal invasion and damage of oral epithelial tissues [25]. Other risk factors are also needed in addition to colonization. In immunocompromised patients, failure to properly treat oral candidosis may lead to chronic persistent infection or invasive candidosis, which may result in necrotizing ulcerative mucositis, suppurative candidosis, inflammatory papillary hyperplasia of hard palate, oropharyngeal and esophageal candidosis, or potentially lethal systemic candidosis, such as *Candida* endocarditis [2,7,26,27]. Patients with candidemia or invasive candidosis may show fever, dysphagia, poor nutrition, slow recovery, or prolonged hospital stay, with a high mortality rate up to 79% [2,4,17]. Although most patients do not become candidemic, over half of all cases of disseminated candidosis or candidemia are undiagnosed at the time of death [2,5,7,16,17]. Therefore, *Candida* in a blood culture should never be viewed as a contaminant and should always prompt searching for the source of fungemia. The incidence of candidemia clearly exceeds that of invasive aspergillosis and mucormycosis and has become an important nosocomial infection in the hospitals [15,16,17].

Oral candidosis is usually easy to treat with good prognosis, but the eradication of fungus to achieve a mycological cure is difficult [2,4,5]. This raises the question whether the recurrence is a second infection or due to persistent *Candida* cells. Clinically, unsuccessful management or recurrence of oral candidosis is not uncommon due to either incorrect diagnosis, missing other pathology, failure to correct the underlying predisposing factors, or improper prescription of antifungal agents, such as recurring use of nystatin oral suspension, inadequate treatment duration, and decrease in drug susceptibility or increase of azole-resistant strains with time [2,7,16,20]. Among these pit falls, the elimination of identifiable predisposing factors is considered the top priority in preventing the recurrence or treatment failure of oral candidosis.

Historically, the aphorism “disease of the diseased” has been given to oral candidosis, and suspicion should therefore be raised when patients have oral candidosis [1,2,5]. It can be a mirror of underlying systemic disease, such as HIV/AIDS, nutritional deficiency, anemia, diabetes mellitus, Cushing’s syndrome, leukopenia, hypothyroidism, end stage of renal or liver diseases, and hematologic malignancies of leukemia, multiple myeloma, or lymphoma (Figure 1). Other oral white lesions, including lichen planus, lichenoid reaction, chronic graft-versus-host disease, leukoplakia, and squamous cell carcinoma, may affect the efficiency of the mucosa barrier and then become superinfected with oral candidosis [4,5,6,7]. These underlying risk factors are usually more important than oral candidosis itself.

Based on our previous studies and over 30 years of oral medicine practice in a medical center [28,29,30,31,32], this review describes the pathophysiology of oral candidosis and provides best practice for diagnosis, practical classification, and successful management of patients with this condition.

## 2. Interplay of Host Defenses and *Candida* Virulence Factors in Oral Candidosis

Systemic host factors are generally considered to be the critical etiology in the development of oral candidal infection, and *Candida* species are strictly opportunistic pathogens that cause a disease state when the host defenses are inadequate [1,2,26,27]. However, this concept has been modified because local host factors, such as oral epithelial abnormalities, mucosa barrier change, impaired salivary gland function, inhaled steroid, wearing dentures, heavy smoking, and a carbohydrate-rich diet, also predispose to oral candidosis (Table 1) [1,2,4,26,33,34]. Oral epithelial cells are the first line of defense against *Candida* infection, functioning as a physical barrier. For a healthy individual, the superficial mucosa protects the host against *Candida* because constant desquamation of oral mucosa takes place at a rate greater than candidal growth. Saliva flow rate and its quantity and quality affect microbial clearance in the oral cavity. Saliva provides many components of adaptive and innate immune response crucial for local host defenses. Saliva contains antifungal factors, such as lysozyme, lactoperoxidase, lactoferrin, and histidine-rich polypeptides, which help to keep the oral *Candida* populations under control [2,5,7]. The elevated levels of IgA, secretory IgA (sIgA), and lysozyme demonstrated in patients with *Candida*-associated denture stomatitis may play a protective role [5,26]. Salivary IgA can also enhance the non-specific antimicrobial effect of lactoperoxidase system [26].

Oral candidosis usually occurs in neonates and the elderly. The immature immune mechanism of newborns and the reduced immune defense in the elderly are the major etiological factors in the pathogenesis of oral thrush [4,5]. However, other co-factors are also important, such as poor oral hygiene, smoking, and dentures. Cigarette smoking can induce increased epithelial keratinization, reduction in salivary IgA levels, and possible depression of polymorphonuclear leukocyte function that may allow *Candida* overgrowth and colonization [7,26,27]. Oral carriage of *Candida* species is minimal in individuals who are totally edentulous prior to insertion of dentures, but that carriage rate tends to increase shortly after a denture is worn [5]. Prolonged denture wearing, poor denture hygiene, and mucosal trauma are important local factors that contribute to developing *Candida*-associated denture stomatitis. The micro-environment of the denture-bearing palatal mucosa is of low oxygen, largely devoid of saliva, and is of low acidic pH, which can promote hydrolytic enzyme activity of *Candida* virulence, enabling invasion of host tissues and avoidance of host defense mechanisms [1,5,35,36]. In addition, oral candidal colonization and candidosis obviously increase iatrogenic factors, such as widespread use of broad-spectrum antibiotics, immunosuppressants, cytotoxic agents, and antipsychotic and antidepressant medications, corticosteroid, inhaled steroids, or irradiation effect [1,2,7,26].

The transition of *Candida* species from the innocuous commensals into disease-causing, opportunistic pathogens is associated with a complex array of interaction between the virulence attributes of the organism and host defenses [1,5,26,33,34,35,36,37,38,39,40,41,42,43,44,45,46,47,48,49,50,51,52,53]. Candidal adhesion to oral mucosa is the essential step in the process of colonization and infection [7,33,34,35,36,37,38,39,40,41,42,43,44,45,46]. The process of adhesion is complex and multifactorial. It is related to interplay of yeast cells, host cells and various environmental factors, such as saliva, sugars, and pH [1,5,7,33,34,35,36,37,38,39,40,41,42,43,44,45,46]. *Candida* is a so-called perfect pathogen that has a remarkable ability to detect alterations in its environment and to respond appropriately by a phenotypic switch of its cell physiology and morphology from yeast to hyphal or pseudohyphal forms that facilitate epithelial penetration [39]. The hyphal cells have one principal property of directional growth (thigmotropism), allowing the fungus to actively invade intercellular junctions. In addition to active penetration, *C. albicans* can invade the host cells by utilizing endocytosis, a passive fungal-induced but host-cell-driven process whereby lytic enzymes and invasins expressed on hyphae bind to and degrade E-cadherin and other inter-epithelial cell junctional proteins, enabling the organisms to be taken up into the epithelial cells [1,34,35,36,38,39,40,41]. *C. albicans* cells also possess a fibrillar layer that is located in the most external surface of the cell wall. These architectures can improve adherence, offer resistance to be scratched off, and also resist saliva flushing and phagocytosis [5,7]. *C. albicans* in oral candidosis also exhibits high rates of alterations in colony morphology between the white and opaque forms [5,7,42,43]. The high frequency switching modes potentiate the pathogenic mechanisms, including the capacity to (1) invade and proliferate in extremely different body environments, (2) elude the immune system by alterations in the surface antigenicity, (3) promote tissue penetration by production of hydrolytic enzymes, (4) form biofilm, and (5) escape antifungal treatment [1,5,7,33,34,35,36,37,38,39,40,41,42,43,44,45,46,47,48,49,50]. An important feature of the cell surface of yeast *C. albicans* is the presence of receptors for the complement fragments C3. The pathogenicity of *C. albicans* is promoted by binding C3 because the phagocytic uptake of the yeast by human neutrophil is impaired [1,5,7,26]. Furthermore, the co-adhesion of *C. albicans* with bacteria is crucial for its persistence, and a wide range of synergistic interactions with various oral species were observed to enhance colonization in the host [37,46,47]. Both antifungal therapy and adequate oral hygiene practices are therefore required to eliminate oral candidosis.

The growth of *Candida* in saliva or carbohydrate-rich nutrient medium is often accompanied by acid production and a concomitant reduction in pH to very low levels. The pH range conductive for candidal growth is about 3.0–8.0, with the optimal conditions for growth being the pH range 5.1–6.9 [54]. However, *C. albicans* and a few other species can grow well at a pH of less than 2.0 [54]. The low acidic conditions would inhibit bacteria flora but favor candidal growth and adherence to epithelial surfaces [5,35,45,54,55]. These acidic metabolites, such as pyruvates and acetates, possess a powerful cytopathic potential that may contribute to the intense mucosal inflammation [5,7,35,41,42,43,44,45,46,47,48]. Dietary carbohydrates can modulate *C. albicans* biofilm development by affecting its virulence factors and structural features [51].

The major component of a *Candida* cell wall is carbohydrates, which form 80~90% of the dry weight of the cell [5]. A high-carbohydrate diet or poor glycemic control in diabetic patients can promote *Candida* proliferation and increase the production of hydrolytic enzymes, such as secreted aspartyl proteinases (SAPs) and phospholipases (LPs), capable of degrading the basement membrane, extracellular matrix, and epithelial cell junctions that not only enhance yeast invasion of the altered oral epithelium but also offer direct cytotoxicity and induce mucosa inflammation seen in atrophic or erythematous candidosis [1,5,7,35,48,51]. SAPs and LPs can digest and destroy cell membranes, which are made of phospholipids with embedded protein. SAPs also allow *Candida* to evade host defenses by degrading molecules of the host immune system, including antibodies and antimicrobial peptides [5,35,48]. Furthermore, a mouse model study discovered that hyphae-induced epithelial damage was mainly mediated through the secretion of a cytolytic peptide toxin called candidalysin, which was essential for the development of oral candidosis [1,40,49,50]. In vitro studies by Samaranayake et al. showed that using glucose-supplemented saliva mixed with added antibiotics to suppress bacterial growth resulted in a rapid decline in pH from 7.5 to 3.2 over a 48-h period, accompanied by yeast growth [5,55,56]. These results suggest that sugar may promote yeast growth [5,51,52,55,56,57]. For example, some young people nowadays may not like to drink pure water but drink soda and beverages that contain high amounts of sugar and favor oral fungus growth. Clinically, many young individuals developed angular cheilitis, atrophic glossitis, diffuse erythematous candidosis, and also rampant cervical caries (“soda teeth”) that alerted the present researcher to the perception of diabetes mellitus and iron deficiency (ID) in these individuals (Figure 1). Studies demonstrate that saliva from patients with ID or diabetes can promote the growth of *Candida* in vitro [7,8,51,52,53]. In vivo susceptibility analysis of antifungal agents combined with glucose in infected diabetic mice showed that an increase in the concentration of glucose also decreased the activity of antifungal agents [52]. Oral candidosis can therefore affect anyone if certain illnesses, stress, dentures, diet, or medications disturb local or systemic immune balance. *Candida* can participate not just passively but actively in the process of pathogenesis to establish oral candidosis. *Candida* pathogenicity can be facilitated by a number of virulence factors (Table 2).

## 3. Iron Deficiency and Immunosuppression in Oral Candidosis

Patients were diagnosed as having iron deficiency anemia (IDA) when men had hemoglobin (Hb) < 13 g/dL, women had Hb < 12 g/dL, and all of them had serum iron level < 60 mg/dL, according to the World Health Organization criteria [58]. ID is further characterized by a low serum ferritin and low transferrin saturation (Fe/TIBC < 16%) or a high total iron-binding capacity (TIBC) [28,58]. IDA is described classically as an anemia of being microcytic, hypochromic, and elevation of red-cell distribution width (RDW) [28,30,58]. Diagnosis of ID, with or without anemia, is not always easy. ID tends to develop slowly, adaptation occurs, and patients often tolerate their symptoms and assume these are normal. They become aware of an improvement only when the symptoms disappear [28]. Several investigators have noticed an association between ID and oral candidosis [14,57,58,59,60,61,62,63,64,65,66,67,68,69,70,71,72,73,74,75,76,77,78]. A high prevalence of ID in patients with diffuse chronic mucocutaneous candidosis (CMC) has been identified, and vitamin deficiencies may also occur [5,59]. CMC is a group of disorders that typically present during childhood and is characterized by recurrent and persistent candidal infections involving the skin, nails, and oral and genital mucosa. CMC is probably the most difficult of the candidal disorders to eradicate; however, correction of iron or vitamin deficiency can occasionally be of benefit [5,57,59]. Higgs and Wells reported that 23 of 31 patients with CMC were iron deficient, and iron therapy alone produced significant improvement in 9 out of 11 of these patients [59]. Clinically, it is unlikely that a single nutrient deficiency is the only factor responsible for chronic oral candidosis, but ID can induce persistent fungal infection, which is difficult to eradicate as long as the deficiency remains [14,28,60,61,62,63].

My previous study reported the first perception of ID or ID anemia (IDA) from oral mucosa changes in 64 patients who demonstrated high incidences of oral candidosis (85%). This resulted in a variety of clinical forms, including angular cheilitis (63%), atrophic glossitis (59%), pseudomembranous candidosis (44%), erythematous candidosis (41%), median rhomboid glossitis (5%), chronic mucocutaneous candidosis (5%), papillary hyperplastic candidosis (3%), and cheilocandidosis (3%). ID glossitis with different degrees of papillary atrophy was often accompanied by pseudomembranous or erythematous candidosis [28]. All patients (50 IDA and 14 ID without anemia) showed relatively few symptoms except fatigue. Nevertheless, oral mucosa alterations appeared. ID was sufficient to promote oral manifestations even in the absence of anemia. Furthermore, colorectal cancer was found in 12.5% of IDA patients older than 65 years [28]. Investigating the origin of IDA is therefore essential because it may be the first sign of a malignant disease [28,62,63,64].

The possibility of persistent candidal infection in ID may be due to impaired cellular immunity [59,60,61,62,63,64,65,66,67,68,69,70,71,72,73,74,75,76,77,78]. However, impaired lymphocyte function cannot entirely explain the mouth lesions and growth of *Candida* in saliva since it is equally depressed in patients with and without mouth lesions. Therefore, local factors, such as the effects of a lack of iron on oral flora change and epithelial abnormalities, may be important [14,28,74,75,76,77,78]. Iron is essential for the growth of all cells. A wide range of non-erythroid alterations of ID, including oral epithelial abnormalities, atrophic gastritis, and thin and brittle nails, such as koilonychias, may occur early before significant changes in red cell morphology or hemoglobin (Hb) level are noted [14,72,73,74,75,76,77,78]. Oral epithelial abnormalities are among the most frequent and important changes that provide a suitable environment for candidal growth responsible for oral candidosis [14,28,65]. Many studies have reported a highly significant reduction in the total epithelial thickness, particularly the thickness of the maturation compartment, and low iron-containing enzyme levels in the buccal epithelium of iron-deficient patients [14,68]. Continued ID leads to reduced hemoglobin levels that carry insufficient oxygen to oral mucosa and finally result in mucosal atrophy [28,68]. ID can subsequently cause lower immunity to infection because of the impaired cellular immunity, deficient bactericidal activity of polymorphonuclear leukocytes, inadequate antibody response, and epithelial abnormality [14,28,70,71,72,73,74,75,76,77,78]. These indicate that ID either acting locally or via systemic mechanisms could be implicated in the pathogenesis of oral candidosis. It also provides new insight that iron therapy could bring benefit in the management of oral candidosis.

In my prior study, most ID patients subjectively felt a better sense of well-being within a couple of weeks of iron therapy [28]. Oral stomatitis resolved completely within 1–2 months. However, continuing iron replenishment for 3–6 months was needed to return iron stores back to normal. Antifungal therapy with nystatin tablets was provided to patients for 2–4 weeks, which resulted in much benefit when oral candidosis existed. Some patients complained about recurrent stomatitis due to inadequate iron absorption several years later and again required iron replenishment. This implies that the final resolution of oral candidosis is by a host defense system, and long-term diet therapy is important for IDA patients [28].

Another study by Lai et al. reported two thymoma patients associated with myasthenia gravis who suffered from recurrent oral candidosis for many years after a thymectomy and chemoradiotherapy [29]. There was an initial good response to conventional antifungal therapy, which later became refractory. Lymphocyte subset quantitation showed a T-cell deficiency and a decreased CD4/CD8 ratio. The thymus gland is the master gland of immunity and the cornerstone for modulation of T-lymphocyte maturation. Levamisole, an immunomodulator, or an immunopotentiating drug were added as adjunctive therapy in combination with oral nystatin treatment. Oral candidosis responded favorably, and substantial relief was obtained with a concurrent increase in T cells and the CD4/CD8 ratio [29]. These findings clearly demonstrate a significant role of cell-mediated immunity in oral candidosis and that final eradication of infection is dependent on the host defense.

The use of CD4 cell count as a marker of clinical disease progression is well established in HIV infection [17,79,80,81,82]. HIV care providers often monitor a patient’s CD4 count to evaluate the effectiveness of an HIV regimen and to determine whether the count has fallen to a level at which an individual might be at risk for certain opportunistic infections. The location and intensity of candidosis in HIV patients are closely associated with a low CD4 count trend. As the CD4 count progressively decreases, the risk of opportunistic candidal infections increases. When the CD4 count is less than 400 cells/mm^3^, there is an increased risk of oral and vaginal candidosis; below 200 cells/mm^3^, there is risk of oropharyngeal candidosis; and when the CD4 count is less than 30 cells/mm^3^, there is an increased risk of esophageal candidosis or systemic candidemia [80]. In this clinical situation, continuous azole antifungals, particularly fluconazole, often led to treatment failure and increased antifungal resistance. However, the highly active antiretroviral therapy (HAART) can control oropharyngeal candidosis with an indirect effect on the CD4 count. Guidelines of HIV management from the Infectious Diseases Society of America (IDSA) suggest that HAART can reduce recurrent oropharyngeal candidosis, and chronic prophylactic antifungal therapy is usually unnecessary [79,80,81,82]. However, certain predisposing factors are quite difficult to eradicate, if not impossible, which requires prophylactic antifungal therapy. Prophylaxis with intermittent prescription of fluconazole three times weekly is recommended when a CD4 count less than 200 cells/mm^3^ develops because nystatin often invites a poor response and rapid relapse of oral candidosis in HIV patients [17,79,80,81,82]. However, the optimal duration of prophylaxis remains known. The high frequency of oral *Candida* carriage and candidosis among HIV/AIDS patients emphasizes that a fully-functional immune system is needed to prevent candidosis.

Clinical research evidence suggests that *Candida* infection is an almost universal finding in patients with severe immunodeficiency of the T-cell type. However, it is rarely seen in patients with B-cell defects in the absence of concomitant T-cell defects [83,84,85,86]. It implies a critical role for cell-mediated factors even at mucosal surfaces. The observation was confirmed in most models of chronic oral candidal infection in immunodeficient mice that lacked T lymphocytes; moreover, complete resolution of the infection in mice was found when they were reconstituted with functional CD4+ T cells [1,83,84,85,86]. All these appear to be strong evidence for the role of CD4 cells in protection against systemic candidosis as well as an important immunoregulatory role for neutrophils. A key component of defense against *Candida*, neutrophils are characteristically seen in the epithelium around hyphae. Interleukin (IL)-17-producing cells appear to play a key role in their recruitment. The source of the IL-17 may be CD4+ T helper 17 (Th17) cells, a view supported by the observations that CD4-depleted HIV/AIDS patients, and Th17-deficient patients are very susceptible to oral candidosis [84,86]. Released cytokines (IL-17 and IL-22) at the site of infection recruit neutrophils, amplifying the secretion of proinflammatory cytokines and chemokines [1]. The Th17-type adaptive immune response is mainly involved in mucosal host defenses, controlling initial *Candida* growth and inhibiting subsequent tissue invasion [1,83]. IL-17 and IL-22 directly influence epidermal immunity and have demonstrated to be associated with protection in many experimental and human *C. albicans* infections [1,84,85,86].

CMC is an infectious phenotype in patients with inherited or acquired T-cell deficiency. The inherited form of CMC is characterized by primary immune deficiencies associated with impaired IL-17 and IL-22 responses [84,85,86]. Several monogenetic defects have been identified to cause CMC, but the most frequent CMC form is a result of heterozygous autosomal dominant gain-of-function mutations in signal transducer and activator of transcription (STAT) 1 [86]. There are also syndromic CMC entities, such as autosomal dominant hyper-IgE syndrome (HIES) caused by impaired STAT3. Despite differences in the clinical presentation of STAT1-CMC and STAT3-HIES, both diseases show comparable reduced Th17 and Th22 immunity and deficient IL-17 secretion, leading to insufficient recruitment of neutrophils from the bloodstream and failure in containing fungal growth on the mucosa [1,85,86]. The induction of Th17- and Th22-cell differentiation was possible in STAT1-CMC patients to a variable degree in vitro, as shown by an increase in cell frequency, but it was impossible to overcome Th17 and Th22 defects in STAT3-HIES patients. The important difference in the two patient groups suggested a strong T-cell receptor engagement as a therapeutic strategy to improve antifungal responses in STAT1-CMC [86]. Early-onset diffuse CMC is the most severe form and is usually resistant to treatment; even the infection remains relatively superficial [5,84,85,86]. Many attempts to correct inherited immune defects in CMC patients by immunoregulators of levamisole, thymosin, and fetal thymic tissue or infusion of leukocytes only produce transient effects [2,66,85]. Surgery or cryotherapy for the removal of persistent oral lesions has been attempted, but the long-term benefits of these procedures are unreliable [5,59,85]. These findings clearly demonstrate a significant role of cell-mediated immunity defects in the pathogenesis of oral candidosis.

## 4. Diagnosis of Oral Candidosis

Clinically, a variety of oral candidosis, namely pseudomembranous candidosis, erythematous candidosis, angular cheilitis, atrophic glossitis, median rhomboid glossitis (MRG), denture stomatitis, cheilocandidosis, mucocutaneous candidosis, hyperplastic candidosis, and oropharyngeal candidosis, or rare suppurative candidosis, can be identified [5,28]. Oral candidosis often presents multifocal and mixed forms that are usually asymptomatic or present themselves as soreness, an unpleasant taste, and burning sensation. It may last for weeks, months, or even years, especially in patients with diffuse CMC.

The diagnosis of any form of oral candidosis is essentially clinical and based on straightforward recognition of the lesion. It is usually not necessary to perform a biopsy except for in the case of candida leukoplakia [28,87,88]. The response to antifungals indicates that oral candidosis is the etiology. Microbiological studies are required when there are diagnostic doubts, resistance to antifungal drugs, or when the antifungal drug dosage needs adjustment, as in immune deficient patients [5,87,88]. Diagnosis can be confirmed with microbiology examination either by staining a smear from the affected area with periodic acid-Schiff (PAS) stain or Gomori methenamine silver (GMS) stain or by culturing a swab from an oral rinse using Sabouraud’s agar. The specimen should be collected from an active fresh lesion; old lesions often do not contain viable organisms [5,7,88]. However, the detection of *Candida* in the oral cavity is not always indicative of infection since it is a common commensal organism in the mouth. The correlation between laboratory findings and clinical manifestations remains the cornerstone in the diagnosis of oral candidosis. The effectiveness of antifungal medication aids in diagnosis. Serological tests, phenotyping, or genetic molecular methods can be useful for systemic fungal infections and have both diagnostic and prognostic values. A number of simplified, rapid, and sensitive commercial kits for identifying *Candida* species based on either the detection of yeast growth in various substrates or the observation of an antigen-antibody reaction are also available [5,7,89]. Rapid yeast identification systems, such as API 20C and ID 32C, appear to be the most widely used [5,89]. However, these are mainly based on biochemical tests and/or color interpretation, often only covering a narrow range of species and requiring a subsequent 24–72 h culture to obtain an identification. The newer technique of matrix-assisted laser desorption ionization-time of flight (MALDI-TOF) mass spectrometry has the advantage of reducing culture time to 24 h and identification of *Candida* species to within minutes [90,91]. MALDI-TOF mass spectrometry has proved to be a rapid and reliable method for identification of *Candida* strains in the clinical laboratory.

Systemic factors affecting the pathogenesis of oral candidosis often include the immune status and nutrition factors of the host. Altered immune functions, like severe leukopenia or lymphopenia, are often the most critical factors predisposing to fungal infection [4]. Altered nutritional statuses, including hematinic deficiencies or diets rich in carbohydrates, act in concert with a number of co-factors [5,7,28,57]. My previous studies have demonstrated that the value of hematological studies in the investigation of patients with unexplainable sore mouth is important, especially in those who have no other obvious causes [28,30]. The hematological investigations consist of complete blood count, (CBC) including hemoglobin (Hb), hematocrit (Hct), mean corpuscular volume (MCV), and red cells distribution width (RDW), measured as standard deviation (SD). The proposed classification of anemia by MCV and RDW of Bessman’s study can be used [92]. The assessments of serum level of ferritin, iron, total iron binding capacity (TIBC), serum B12, and folate should be made when iron and vitamin deficiency are highly suspected [28,30,92,93,94,95,96,97]. Further screening of HIV infection, diabetes, liver or kidney disease, and malignances has to be undertaken when patient’s history and physical examinations suggest the possible pathology. Each patient should be consulted by a hematologist or medical doctors for further evaluation and treatment.

## 5. Classification of Oral Candidosis

By tradition, the most frequently adopted classification of oral candidosis as proposed by Lehner in 1966 divided oral candidal infections into acute and chronic types of pseudomembranous candidosis and atrophic candidosis. Lehner recognized that chronic oral hyperplastic candidosis was a common manifestation of CMC [5,98]. He suggested further subdivision of the chronic hyperplastic candidosis into four groups based on the localization pattern and on endocrine involvement as (1) chronic oral candidosis, (2) endocrine candidosis syndrome, (3) chronic localized mucocutaneous candidosis, and (4) chronic diffuse candidosis [5,98,99,100,101]. However, the last subdivision of oral hyperplastic candidosis generated some confusion, particularly between CMC and hyperplasia. Moreover, atrophic candidosis has been revised as erythematous candidosis because over 60% of cases demonstrate an increase in the epithelial thickness rather than atrophy [5,28]. Several investigators have noticed that the red color of oral candidosis seems more prevalent than white lesions. Professor Jens Jorgen Pindborg, who was an internationally renowned oral pathologist, stated that when our predecessors christened the fungus *Candida*, which means white, they did not realize that candidosis is most often characterized by a red color [5].

Holmstrup and Bessermann in 1983 suggested a dichotomous classification of oral candidosis that highlighted and divided candidosis confined to oral and perioral tissues into one group as the primary oral candidosis, and other disorders where oral candidosis with concomitant generalized systemic mucocutaneous candidal infection were categorized as secondary oral candidosis [5,99]. The primary oral candidosis lucidly described all known forms of the disease, including acute and chronic types of pseudomembranous candidosis and erythematous candidosis, *Candida*-associated angular cheilitis, and chronic plaque-like or nodular candidosis [5,98,99].

For oral candidosis with concomitant systemic mucocutaneous affections (secondary oral candidosis), the common classification of CMC as proposed by Odds in 1988 included six categories as (1) CMC with endocrinopathy, (2) familial CMC with endocrinopathy, (3) familial CMC without endocrinopathy, (4) idiopathic CMC with juvenile onset, (5) idiopathic CMC with mature onset over the age of 20 years, and (6) CMC associated with thymoma [5,98,99,100,101]. However, the etiology of CMC is multifaceted, involving the interplay of genetic, hormonal, and immunological factors. There are always CMC patients who cannot easily be confined to a single subdivision; moreover, the vast majority of CMC cases nowadays are more likely to occur in adult patients with acquired diseases related to immunosuppression, such as IDA, diabetes mellitus, neutropenia, endocrinopathy, or HIV/AIDS. The protean nature of CMC and associated diseases makes it almost impossible to compartmentalize the many and varied manifestations of the disease into a rigid system of classification without difficulty and confusion [5,101]. The chronic oral thrush in diffuse CMC is initially pseudomembranous and then soon gives rise to chronic hyperplastic lesions that present as flat or raised white, adherent plaques affecting any oral mucosal surface. These thick white plaques cannot be rubbed off; even the infection remains relatively superficial (Figure 2).

*Angular cheilitis and atrophic glossitis* remain the most common oral manifestations of hematinic deficiencies warranting further investigation with blood tests [28,30,61]. Angular cheilitis is considered a component of chronic multifocal candidosis. The presentation may be unilateral but is more often bilateral. However, angular cheilitis could either be due to infective or non-infective etiology. A significant proportion of the lesion is due to candidal infection or synergistic infection with *Candida* species and *Staphylococcus aureus* that has been shown by microbiological studies [5,28,98]. Atrophic glossitis is well-known as acute atrophic candidosis or *Candida* glossitis. *Candida* glossitis can present as pseudomembranous or erythematous candidosis [28,102].

*Median rhomboid glossitis* (MRG), which often occurs in anemic or diabetic patients, is associated with candidal infection and is not a developmental anomaly, as pediatric cases are seldom encountered [28,103,104]. MRG is characterized by central papillary atrophy that presents as a central elliptical or rhomboid area of atrophy and erythema of the midline posterior tongue dorsum, anterior to the circumvallate papillae. High levels of *Candida* can be recovered from these lesions, and a biopsy from this lesion usually yields *Candida* in over 85% of cases [104]. MRG as a variant of erythematous candidosis can be confirmed by a complete response to antifungal therapy (Figure 3) [28,103,104]. However, complete regeneration of the papilla in MRG may not occur in long-standing lesions. However, excision of the lesion is usually unnecessary.

*Candida-associated denture stomatitis* is often characterized by chronic erythema and edema of a part or the entire mucosa of the palate and the alveolar ridges that contact with the dentures. However, denture stomatitis may show mixed erythematous and pseudomembranous candidosis accompanied by angular cheilitis (Figure 4). Denture stomatitis is seen in up to 75% of denture wearers, and often, there are no clinical symptoms [1,33,105]. Frictional irritation by ill-fitting dentures can harm the mucosal barrier, allowing infiltration of *Candida* into the tissue causing infection [5,33]. The acrylic denture base and denture-relining materials act as a chronic reservoir allowing continuous seeding of *Candida* onto the palatal tissue; this in turn elicits a robust local inflammatory response as tissue erythema and hyperplasia [5,33,105,106]. This condition is considered a classic *Candida* biofilm-associated infection [5,33]. Three types of denture stomatitis proposed by Newton in 1962 can be distinguished as (1) a localized, simple inflammation or a pinpoint hyperemia; (2) a diffuse erythematous or generalized simple type involving the entire denture-covered mucosa; and (3) a granular type (inflammatory papillary hyperplasia) involving the central part of the hard palate and the alveolar ridges [5,90,91,92]. The granular type is often seen in association with the generalized simple type. However, the granular type or inflammatory papillary hyperplastic candidosis can exist in immunodeficient patients, such as ID or systemic lupus erythematosus cases, without dentures (Figure 5) [5,29,105,106,107].

*Candidal leukoplakia* or *chronic hyperplastic candidosis* is either homogenous or nodular (speckled) and tends to be asymptomatic. However, there is not agreement whether this lesion should be named *Candida*-leukoplakia or hyperplastic candidosis and whether *Candida* infection is the cause of leukoplakia or is an infection superimposed in a pre-existing lesion. It can only be confirmed by the presence of *Candida* hyphae in the lesion and complete resolution after empirical antifungal therapy [5,100]. A biopsy is required because a higher malignant transformation rate of 9–40% has been reported in *Candida*-infected speckled leukoplakia when compared with the 2–6% risk of malignant transformation cited for homogenous leukoplakia in general [5,7,100]. Folic acid deficiency or ID was detected in 33% of patients with candida leukoplakia [5,7]. However, only a minority of patients with candida leukoplakia have associated medical conditions. Those reported included malabsorption syndrome, diabetes mellitus, and asthma treated with steroid inhalers [5,7,100].

*Oropharyngeal candidosis* may show lesions involving the back third of the tongue, the soft palate, the side and back walls of the throat, and the tonsils. It has been considered a reliable marker of immunosuppression or HIV infection associated with CD4+ T-lymphocytes count less than 200 cells/ mm^3^ [18,19,80,81,108,109,110]. The novel coronavirus (SARS-CoV-2) is armed by special abilities to spread and dysregulate the immune mechanisms. The occurrence of oropharyngeal candidosis in critically ill COVID-19 patients was observed [108]. Oropharyngeal candidosis may be a risk factor for systemic candidosis, particularly in critically ill patients who have fever despite antibacterial therapy and multiple interrelated risk factors, such as severe neutropenia, lymphopenia, uremia, organ transplant on immunosuppressants, or having invasive medical device and total parenteral nutrition (Figure 6) [80,83,108,109,110]. However, the aggressive nature of systemic candidosis may be due to prior incorrect diagnosis and erroneous therapy. About one-third of febrile neutropenic patients who do not respond to one-week course of antibiotic therapy have systemic fungal infections. Early empiric antifungal therapy is crucial because it can obviously decrease mortality. Furthermore, blood or wound culture of fungus for species identification may be necessary [108,109,110].

*Suppurative candidosis* is a rare clinical entity that may show necrotizing ulcerative mucositis, submucosal abscess, cellulitis-like dermis infections, or osteomyelitis of jawbones [111,112]. Diagnosis is often delayed until it fails to respond to conventional antibiotic treatment, and the results of repeated cultures show the causative organism to be *Candida.* Treatment consists of surgical drainage, debridement, and use of azole antifungal drugs (Figure 7).

*Juxtavermillion candidosis* is a chronic, self-limiting candidal infection of the lip vermilion and juxtavermilion skin in young children or adults that is apparently triggered by habitual lip licking or minor trauma [113,114]. These infections typically appeared as erythematous, pruritic, yellow crusting plaques of the juxtavermilion skin with or without desquamation of vermilion surfaces (Figure 8). Evidence of intraoral candidosis, especially loss of filiform papillae of the tongue, may present in several cases. The disorder mimics the early stage of CMC but remains within a few millimeters of the mucocutaneous junction, and the affected individuals appear to be immune competent [5,113,114]. 

Clinically, candidal infection is often the leading cause of chronic oral ulcers after excluding other possible causes, such as trauma, mucocutaneous diseases, syphilis, neoplasm, nutritional deficiency, or drug reaction in critically ill patients. This was based on the present researcher’s clinical experience in a medical center over three decades. When the adherent white spots or plaques are sloughed off, the chronic oral ulcers may show a variety of aspects, such as diffuse erosive mucosa with bleeding, continuous small round ulcers, or a large ulcer covered by a thick shiny plaque with grayish or yellowish color change as an organized biofilm-like material on the infected mucosa (Figure 9). *Candida* biofilm is a structured microbial community that is attached to a surface and surrounded by a self-produced extracellular matrix. The biofilm is usually resistant to most antifungals and difficult to control by conventional nystatin oral suspension [36,48,51,52,53].

In this report, the present researcher suggests a practical classification of oral candidosis based on clinicopathological features and locations (Table 3). Four new entities, such as cheilocandidosis, juxtavermillion candidosis, oropharyngeal candidosis, and rare suppurative oral candidosis, are included for the first time and may be worthy of mention in this classification. Oropharyngeal candidosis and rare suppurative oral candidosis are an ominous sign of invasive candidosis and systemic candidemia [5,108,109,110,111,112]. Cheilocandidosis described by Reade et al. in 1982 and juvenile juxtavermillion candidosis by Bouquot and Fenton in 1988 are often ignored, which manifest around the lips of affected young persons and respond well to antifungal therapy [5,113,114].

Our previous study proved that CMC related to ID or acquired immunosuppression can be a primary candidal infection only restricted to the oral and perioral sites [28]. Not all of the CMC cases, therefore, were secondary infections by *Candida* species with concomitant systemic mucocutaneous candidosis. The prior classification of oral candidosis into either primary or secondary candidal infection may not be appropriate because secondary infection is defined as an infection that occurs either during or immediately following treatment for another infection or disease. It may be caused by the first treatment or by changes in the immune system. Theoretically, oral candidosis occurs commonly as a secondary infection in immunocompromised patients or long-term steroid and broad-spectrum antibiotic users. The dichotomous classification of oral candidosis into primary oral candidosis restricted to the oral and perioral sites and secondary oral candidosis as candida lesion in oral cavity accompanied with systemic CMC may give rise to considerable confusion [2,99]. In the study, CMC is simply divided into congenital or inherited type since childhood and acquired type in later life. The acquired type of CMC can be focal or diffuse. Lu’s new classification of oral candidosis includes all known forms of the disease that may be more practical to use (Table 3). However, a limitation of this reclassification is that it is based on a single clinician’s observations and experience over several decades of practice. Future research in this area would benefit from more clinicians’ opinions to keep this reclassification updated.

## 6. Management for Oral Candidosis Patients

Oral candidosis is a treatable disease, and the prognosis is usually positive in the majority of patients [1,2]. A thorough medical history and appropriate workup are crucial for the successful treatment of patients with oral candidosis. The likelihood of developing oral candidosis can be reduced through the elimination of risk factors and maintenance of efficient oral hygiene. Mechanical means to remove heavy candidal plaques from oral lesions is important and can promote antifungal action and speed healing (Figure 10) [5,7,112,115].

From the first antifungals, nystatin and amphotericin in the late 1950, to the early 21st century, over 60 years, only a small number of antifungal drugs have become available, and most are fungistatic. Nystatin remains the first option for oral candidosis. Nystatin shows poor gastrointestinal absorption and no systemic effect. It acts locally, and the key to working well is by direct contact with the affected tissues with enough time and dosage. Swallowing nystatin tablets or capsules rather than sucking or dissolving them in the mouth is ineffective in treating oral candidosis [2,82,115,116]. Studies reported that nystatin pastille alone or pastille and suspension in combination was more effective than that of suspension alone [28,116]. The flushing effect of saliva or the cleansing action of the oral musculature tends to reduce the drug concentration of suspension to sub-therapeutic levels. Patients should be emphatically informed to hold nystatin capsule in the mouth and suck slowly for a longer time (3–5 min) before swallowing [28]. Clinically, many patients with chronic oral candidosis have been refractory to nystatin suspension for weeks but showed rapid response to nystatin lozenges within one week. Antimycotic treatment duration should be enough. It is recommended to double the time needed for resolution of the clinical signs of infection or prolong antifungal treatment for at least four weeks to obtain a more permanent mycological cure after therapy, but patient’s compliance to topical antifungal agents may be compromised due to taste intolerance to nystatin and the prolonged treatment duration [5,28].

In vitro study showed that chlorhexidine can suppress the antifungal activity of nystatin. Chlorhexidine may interact with nystatin to form a complex that can render both agents ineffective against *Candida*. This indicates that combined oral therapy with nystatin and chlorhexidine should not be carried out [28,117]. For *Candida*-associated denture stomatitis, the irregularities of acrylic denture base and denture-relining materials are a reservoir of *Candida* hyphae and yeast cells. Elimination of yeast cells from the inflamed palatal mucosa without disinfecting the dentures will lead to recurrent infection [1,5,7,28,100]. It is essential to eradicate yeast residing both on the oral mucosa and in the dentures. Patients should be instructed to take the following precautions: (1) removal of dentures during the sucking of nystatin lozenges, (2) wearing the dentures as seldom as possible, (3) efficient oral and denture hygiene, (4) keeping dentures in a disinfectant solution of 0.2% chlorhexidine overnight, and (5) sufficient antifungal treatment duration.

Second-line therapy for oral candidosis is dominated by azole antifungal agents. Early azoles include clotrimazole, miconazole, and ketoconazole, but some disadvantages and side effects limit their use. Clotrimazole and miconazole act locally and have double effect against Gram-positive bacteria and *Candida*. Both can be used to treat angular cheilitis. On the other hand, fungus eradication from a clinical lesion by only applying topical antifungal agents is inadequate; the source of pathogens in *Candida*-associated angular cheilitis is commonly the inside of the mouth [7]. Ketoconazole as the first antifungal of the oral route is highly bound to plasma protein and penetrates poorly into saliva or urine. Clinicians should no longer prescribe ketoconazole as the first-line therapy for any fungal infection, including oral candidosis, due to the poor concentration in the affected oral tissues and the risk of severe hepatoxicity, nephrotoxicity, teratogenic effect, and adverse drug interactions [118,119]. Its use is only beneficial when other effective antifungals are not available, or potential benefits of oral ketoconazole outweigh potential risks.

Triazoles, including fluconazole, itraconazole, voriconazole, and posaconazole, can be used for the treatment of systemic fungal infections, oropharyngeal and esophageal candidosis, or mucocutaneous candidosis and intermittent maintenance therapy in patients who have a weak immune system caused by cancer treatment, bone marrow transplant, or diseases such as AIDS [79,80,81,82,109,120]. Fluconazole is on the World Health Organization’s List of essential medicines and has many distinguishing properties, such as excellent gastrointestinal absorption with a very long serum half-life and convenient once-daily dose for all indications. The pharmacokinetic properties of fluconazole are similar following administration by the intravenous or oral routes. It can achieve strong correlation between concentrations in the serum, saliva, urine, and the affected tissues. The concentration at the site of infection, not the concentration in serum, is the important antifungal activity [118,119,120,121,122,123,124]. Administration of twice the usual daily dose of fluconazole as a loading dose is a pharmacologically rational way to achieve higher steady-state blood concentrations more rapidly. Fluconazole is available as a powder for oral suspension. Clinical cure rates and patient’s compliance for using fluconazole is superior to nystatin suspension for the treatment of oral candidosis [123].

Candidosis often occurs in an immunocompromised patient who tends to take many drugs, but many drugs can interact with azoles. The interaction of azoles and a patient’s drug regimen must be carefully evaluated before the addition or removal of an azole drug. Severe interaction with certain drugs (cisapride, dofetilide, pimozide, quinidine) can lead to QT interval prolongation, life-threatening ventricular tachyarrhythmias, such as torsades de pointes, or toxic epidermal necrolysis (TEN), and drug reaction with eosinophilia and systemic symptoms (DRESS). For patients with poor renal function, the delay in renal clearance of azole antifungal agents will increase serum drug concentration and also cause side effects such as headache, nausea, and liver toxicity [115,118,119,120,121,122,123,124,125].

A wide difference in the antifungal susceptibility testing and drug dosing to fluconazole between *Candida* species exists. The therapeutic drug concentrations within the plaque may not be enough to eradicate the yeasts; moreover, a high risk of selection and enrichment of azole-resistant strains increase after long-term use [82,120,121,122,123,124,125]. Oropharyngeal and esophageal candidosis remain significant causes of morbidity in HIV patients despite the dramatic ability of antiretroviral therapy to reconstitute immunity. Risk factors for azole-resistant oropharyngeal candidosis in AIDS patients, such as a low CD4 count, prolonged or multiple courses of therapy, intermittent therapy, a low daily azole dose (<100 mg/day), and a high total cumulative azole dose, have been identified [79,80,81,82]. For these reasons, *Candida* species identification and susceptibility testing for azole resistance is increasingly used to guide the proper management of recurrent oropharyngeal candidosis in HIV patients. The traditional systemic antifungal of amphotericin B and newer antifungal agent of echinocandin (so-called antifungal penicillin; caspofungin, micafungin, and anidulafungin) can offer a broad spectrum of activity for *Candida* species and second-line therapeutic choice for invasive candidosis [124].

Dentists and physicians should be aware of these above-discussed key points and properly prescribe antifungals in order to successfully manage oral or oropharyngeal candidosis to prevent recurrence and systemic spread. Patient education on using antifungals is also essential.

## 7. Conclusions and Important Suggestions for Management of Oral Candidosis

(1)Maintenance of good oral and denture hygiene is crucial. It is important to remove dentures overnight, use denture cleanser, or make a new denture if an ill-fitting denture with stomatitis exists.(2)Rinsing the mouth after use of an inhaled steroid is helpful to prevent oral candidosis.(3)Glucose promotes yeast growth, and a high-carbohydrate diet enhances its adherence to oral epithelial cells. Limiting their consumption is helpful in the control of oral *Candida* colonization and infection.(4)Removal of heavy candidal plaques or biofilm from oral lesions by mechanical means can improve antifungal action and speed healing.(5)Nystatin tablets are significantly superior to nystatin oral suspension in treating oral candidosis. Swallowing nystatin tablets rather than sucking or dissolving them in the mouth is ineffective to treat oral candidosis.(6)The duration of antifungal treatment should be sufficient or prolonged for at least four weeks to achieve a more permanent mycological cure.(7)Early fluconazole monotherapy or fluconazole combined with nystatin is helpful to treat oropharyngeal candidosis, suppurative candodosis, or *Candida*-related chronic oral ulcers.(8)Attempts to increase CD4 count in patients with HIV/AIDS or thymoma are helpful to treat oral candidosis. HAART can reduce recurrent oropharyngeal candidosis.(9)Underlying predisposing factors should be identified and treated simultaneously as well as monitored regularly.(10)The final eradication of oral candidosis is by host defense system.

## Figures and Tables

**Figure 1 jof-07-00555-f001:**
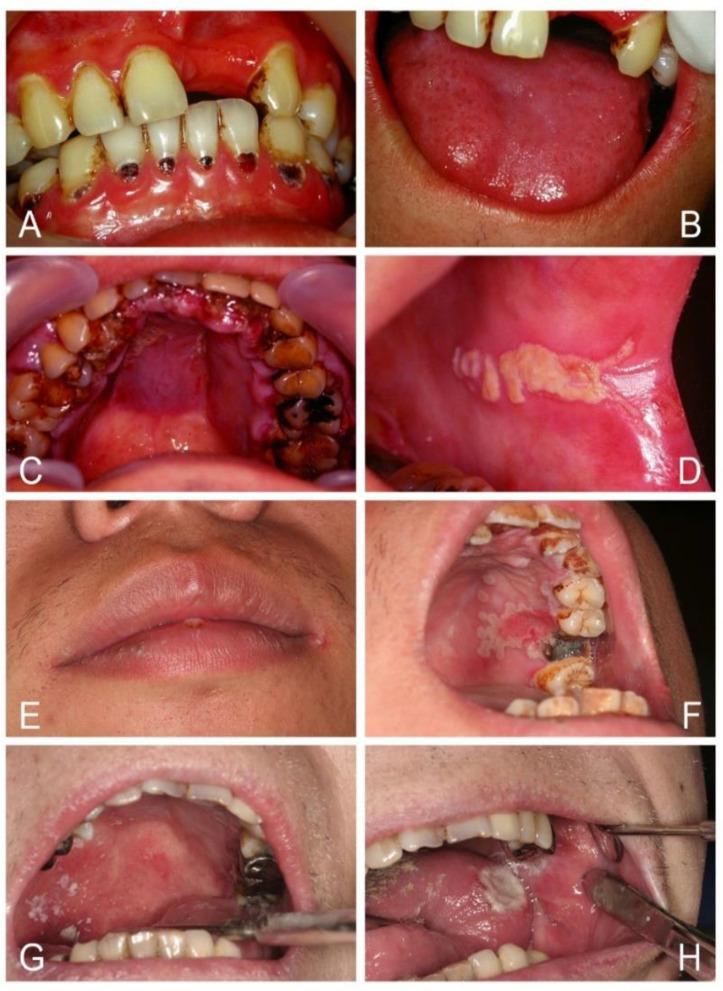
Oral candidosis can be a mark of systemic disease. (**A**,**B**) Erythematous candidosis, angular cheilitis, and rampant cervical caries led to the diagnosis of diabetes mellitus and iron deficiency anemia in a 30-year-old male who drank soda daily instead of water. (**C**,**D**) Pseudomembranous candidosis and diffuse gum hyperplasia with cyanotic change led to the diagnosis of acute myeloid leukemia in a 32-year-old man. (**E**,**F**) Cheilocandidosis and pseudomembranous candidosis around a necrotic ulcer over the palate led to the diagnosis of HIV infection in a 28-year-old man. (**G**,**H**) Pseudomembranous and erythematous candidosis with a necrotic tongue ulcer led to the diagnosis of severe leukopenia in a 33-year-old woman.

**Figure 2 jof-07-00555-f002:**
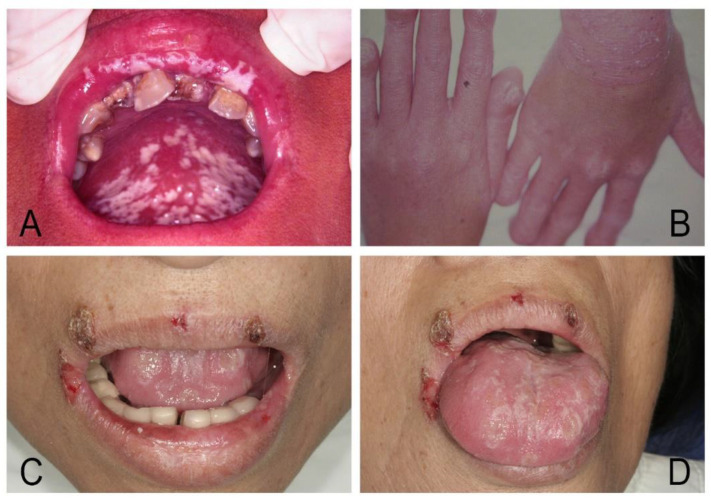
Chronic mucocutaneous candidosis (CMC). (**A**,**B**) Congenital type of CMC in a 14-year-old girl showed disfigured and clubbing nails, crusted rash of skin, and persistent oral thrush that was initially pseudomembranous and then changed to hyperplastic lesions as flat or raised white adherent plaques hard to be rubbed off; even the infection remained relatively superficial. (**C**,**D**) Acquired type of CMC limited to the oral and perioral sites in a 56-year-old woman with iron deficiency anemia.

**Figure 3 jof-07-00555-f003:**
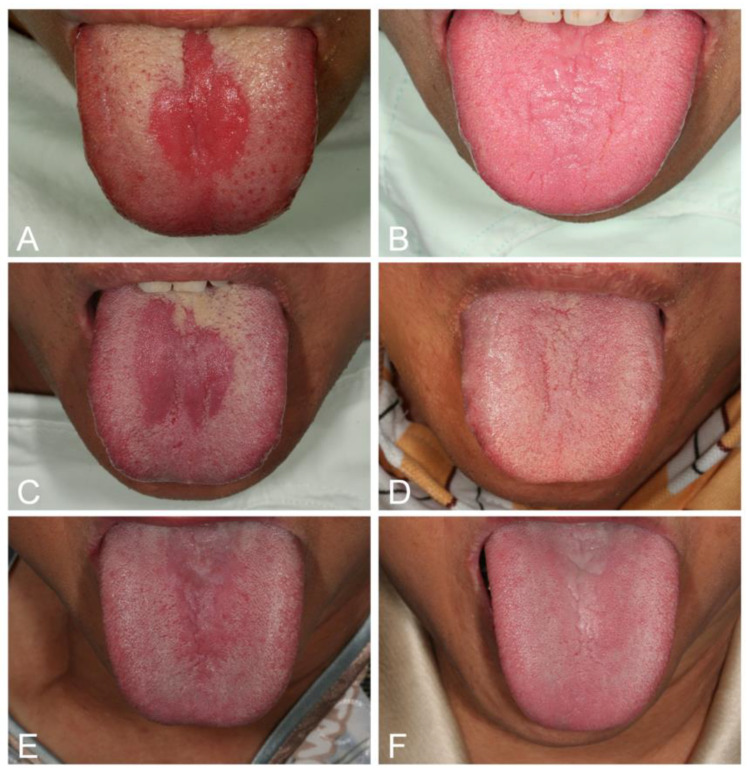
Median rhomboid glossitis (MRG) as a variant of erythematous candidosis (**A**,**C**,**E**) in three women with iron deficiency anemia and one of them also with diabetes. They showed complete response with regeneration of tongue papilla following iron combined with nystatin therapy 2 weeks later (**B**,**D**,**F**).

**Figure 4 jof-07-00555-f004:**
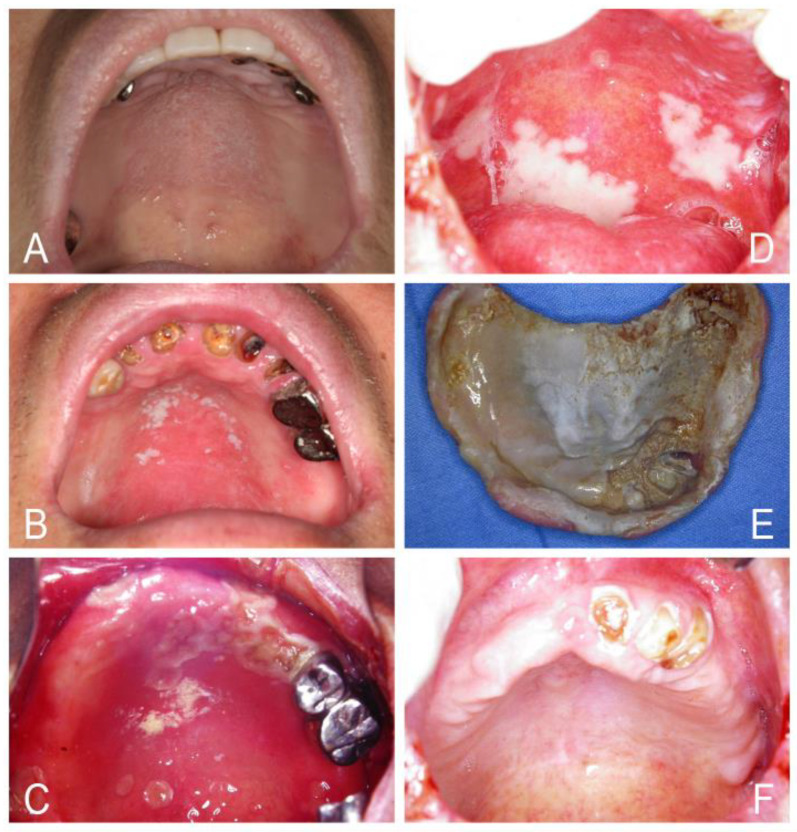
*Candida*-associated denture stomatitis (**A**–**D**) showing a more diffuse erythema involving a part or the entire denture-covered mucosa combined with pseudomembranous candidosis and angular cheilitis in four people who wore dentures for many years. An ill-fitting denture (**E**) showed poor denture hygiene and the denture-relining materials acted as a chronic reservoir allowing continuous seeding of *Candida* onto the palatal tissue of the patient (**D**). Complete resolution of denture stomatitis after antifungal therapy with nystatin tablets, efficient oral and denture hygiene and removal of the denture (**F**).

**Figure 5 jof-07-00555-f005:**
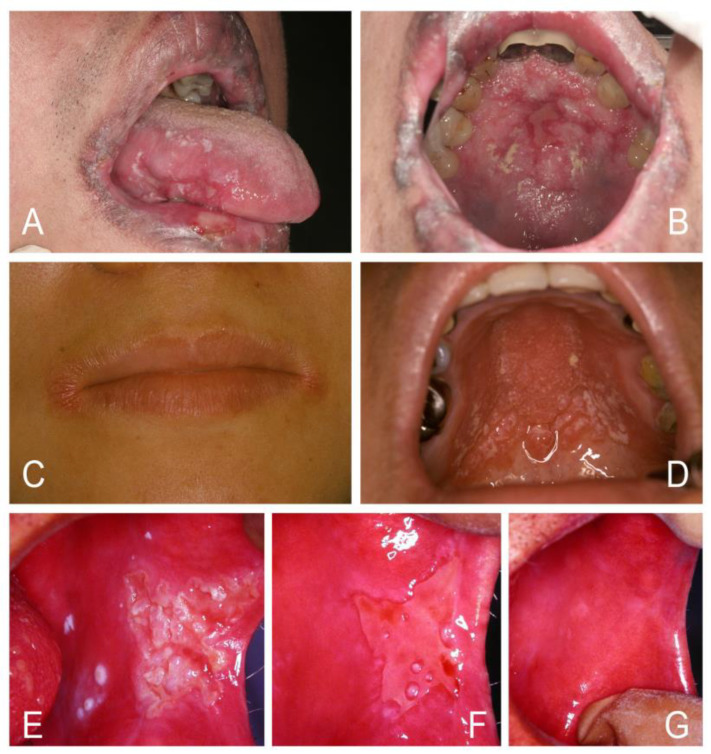
(**A**,**B**) Pseudomembranous candidosis and granular type or inflammatory papillary hyperplastic candidosis of palate and recurrent cheilocandidosis with heavy lip stain in a 35-year-old man showing systemic lupus erythematosus. (**C**,**D**) Angular cheilitis and inflammatory papillary hyperplastic candidosis of palate in a 25-year-old woman showing iron deficiency and moderate anemia. (**E**) Pseudomembranous candidosis and inflammatory papillary hyperplastic candidosis of buccal mucosa in a 50-year-old man with diabetes mellitus who showed dramatic response to nystatin therapy 5 days later (**F**) and complete response to therapy 10 days later (**G**).

**Figure 6 jof-07-00555-f006:**
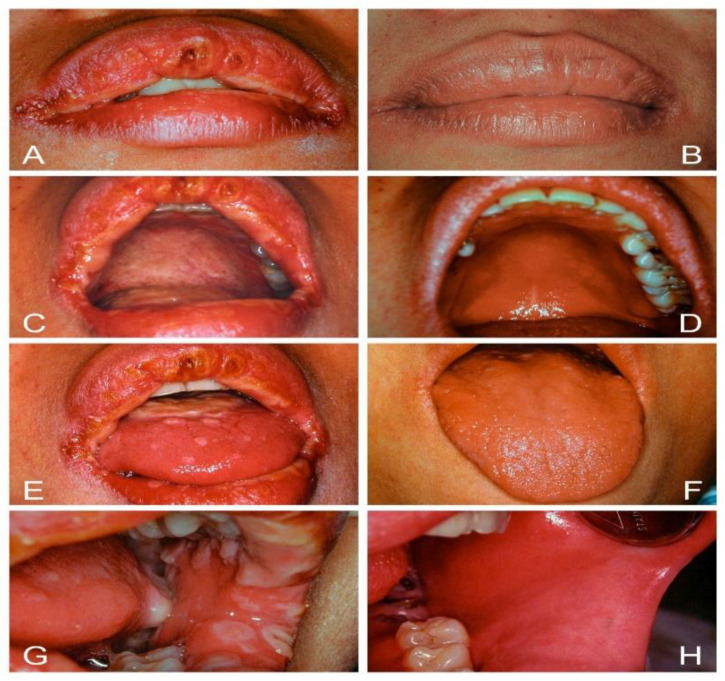
(**A**,**C**,**E**,**G**) Oropharyngeal candidosis resulting in high fever and potential candidemia in a 24-year-old woman with systemic lupus erythematosus and uremia who was refractory to therapy with nystatin suspension, ketoconazole, and vancomycin for one week in a local hospital. (**B**,**D**,**F**,**H**) After prescription with fluconazole combined with nystatin tablets, she showed no fever 2 days later and complete response within one month.

**Figure 7 jof-07-00555-f007:**
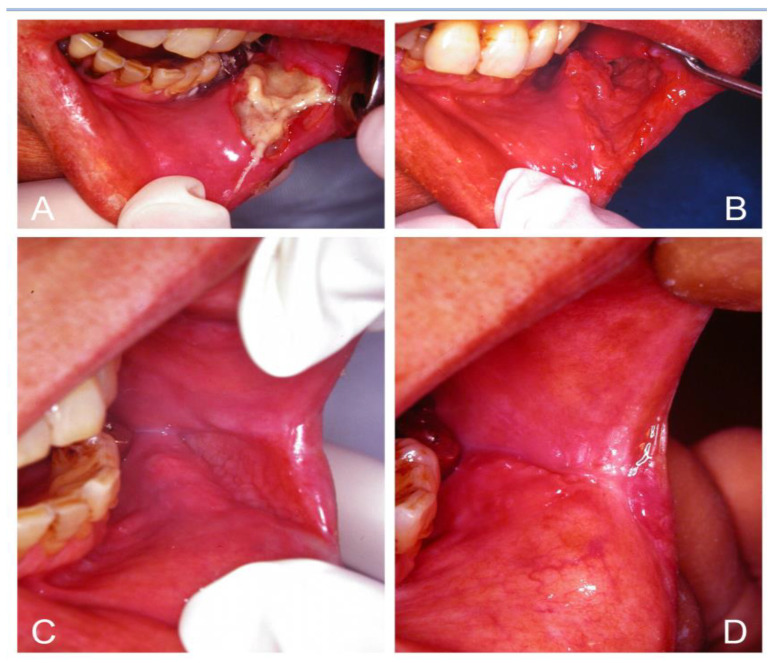
(**A**) Suppurative candidosis resulting in a deep necrotic oral ulcer with facial cellulitis-like dermis infection and fever in a 64-year-old woman with nephrotic syndrome who was refractory to broad-spectrum antibiotic therapy for 5 days. (**B**–**D**) Successful management was achieved by local debridement and prescription with fluconazole 1 week later (**B**), 3 weeks later (**C**), and complete response within 4 weeks (**D**).

**Figure 8 jof-07-00555-f008:**
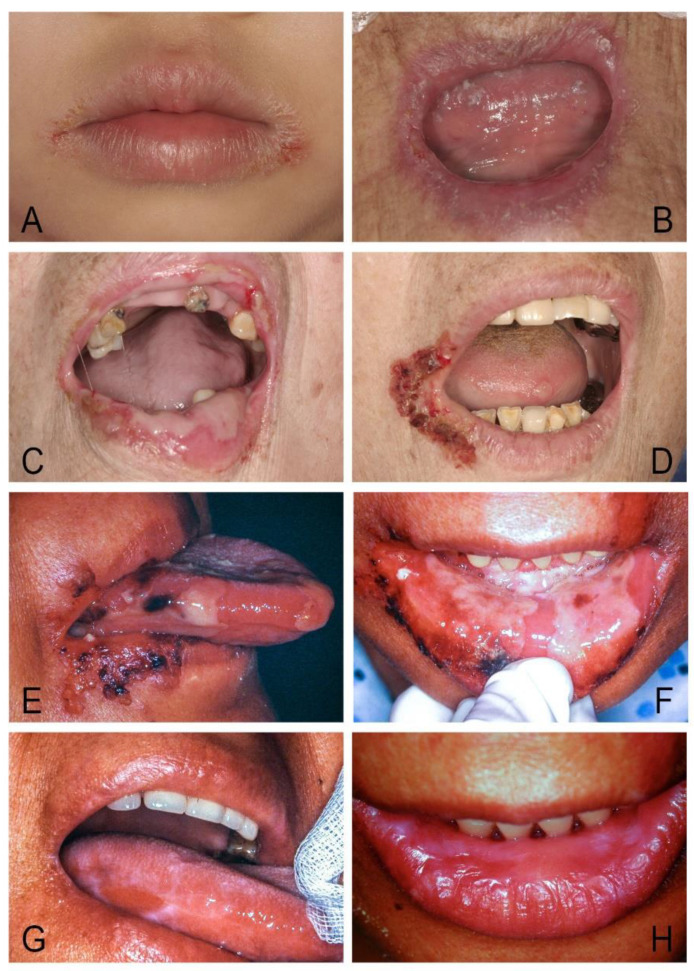
(**A**) Juvenile juxtavermillion candidosis in a healthy 4-year-old girl. (**B**–**D**) Cheilocandidos with intraoral candidosis in three immunocompromised patients. (**E**,**F**) Candidosis-related angular cheilitis and oral ulcers before therapy in a 60-year-old woman with diabetes and uremia. (**G**,**H**) Complete response to nystatin tablets and clotrimazole cream 10 days later.

**Figure 9 jof-07-00555-f009:**
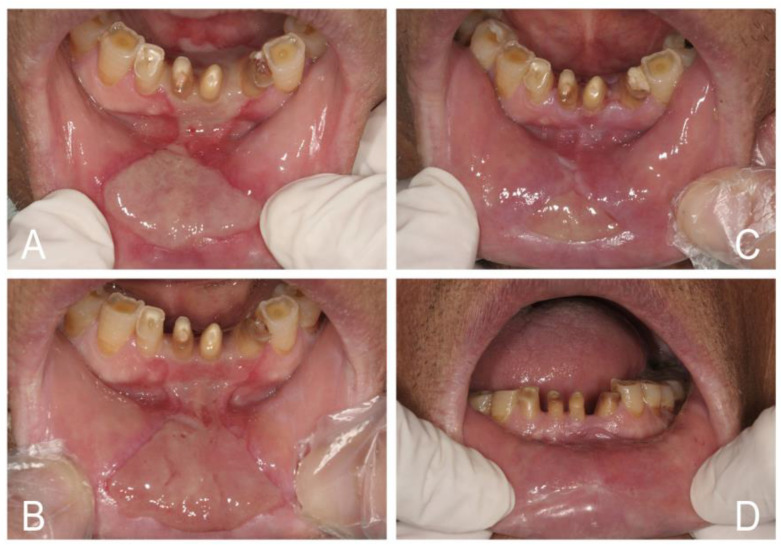
(**A**) The change of chronic oral ulcers associated with oral candidosis covered by a thick shiny organized biofilm with grayish color over lower labial mucosa, anterior gingiva, and tongue tip over 1 year in a critically ill 68-year-old woman with osteoarthritis and Reiter’s syndrome on many medications, including prednisolone. (**B**) Progressive resolution of the lesion was achieved by initial prescription with fluconazole 2 weeks later (**B**), 4 weeks later (**C**), and then changed to nystatin tablets and complete response within 6 weeks (**D**).

**Figure 10 jof-07-00555-f010:**
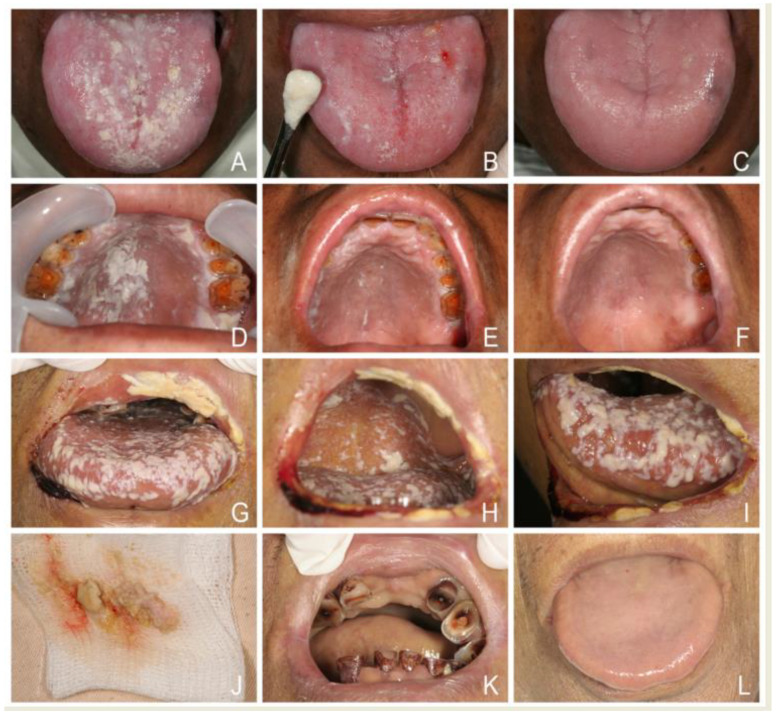
(**A**,**D**) Pseudomembranous candidosis in a 60-year-old man with nasopharyngeal carcinoma after head-and- neck radiotherapy. (**B**,**E**) Using a periosteal elevator to remove heavy candidal plaques from tongue and palatal lesions. (**C**,**F**) The mechanical means promote antifungal action of nystatin tablets and speed healing within one week. (**G**–**I**) Severe oropharyngeal pseudomembranous candidosis showing diffuse lesions as multiple adherent white spots or larger plaques like curdled milk or cottage cheese in a 59-year-old man with liver cirrhosis was refractory to nystatin oral suspension prescribed by a physician for several days. (**J**–**L**) Rapid resolution of oropharyngeal candidosis by mechanical removal of heavy white plaques and prescription with nystatin tablets one week later.

**Table 1 jof-07-00555-t001:** Systemic or local host factors predisposing to oral candidosis.

Systemic Host Factors	Local Host Factors
**Altered physiological status**	**Mucosal barrier alterations**
Infancy/old age	Exogenous epithelial change
**Altered hormonal status**	Trauma
Diabetes	Loss of occlusion
Hypothyroidism/Hypoparathyroidism	Maceration
Cushing’s syndrome	Endogenous epithelial changes
**Altered hematinic or nutritional status**	Atrophy
Iron deficiency	Hyperplasia
Hypovitaminosis, Vit B12, folic acid	Dysplasia/Oral cancer
Malnutrition	**Saliva quantitative changes**
**Altered immune status**	Xerostomia
Defects in cell-mediated immunity	Sjogren’s syndrome
Reduced numbers of phagocytes	Radiotherapy/Cytotoxic therapy
Lymphopenia or leukopenia	**Saliva qualitative changes**
Decreased CD4 count	pH/glucose concentration
Due to infective states/ HIV	**Poor oral or denture hygiene**
**Blood dyscrasias/ malignancies**	**High carbohydrate diet**
**Immunosuppressant/ chemotherapy**	**Heavy smoking/ Betel nut chewing**
**Broad spectrum antibiotics**	**Inhaled steroid**

**Table 2 jof-07-00555-t002:** Potential virulence factors of *Candida* species.

Virulence Factors	Effects
**Adherence**	**Promote retention in the mouth**
Expression of cell surface adhesins	Specific adhesion
Cell surface hydrophobicity	Nonspecific adhesion
**Invasion and destruction of host tissue**	**Enhance pathogenicity**
Hyphae development/thigmotropism	Promote invasion/active penetration
Secret hydrolytic enzymes	Cytotoxicity to oral epithelium
Secret acidic metabolites	Degrade basement membrane/matrix
Endocytosis	Passive penetration of epithelium
**Evasion of host defense**	**Reduce phagocytosis/help retention**
Phenotypic switching	Antigenic modification
Proteolytic degrading immune factors	Destroy sIgA/ antimicrobial peptides
Binding of complement	Antigenic masking
**Synergism with bacteria**	**Promote mixed-species retention**

**Table 3 jof-07-00555-t003:** New classification of oral candidosis by clinicopathological features and locations.

**Acute/Chronic Candidosis**	**Hyperplastic Candidosis**
Pseudomembranous candidosis	Plaque-like
Erythematous candidosis	Nodular-like
**Candida-associated lesions**	Granular or papillary-like *
Denture stomatitis	**Keratinized primary lesions superinfected with Candida**
Localized simple inflammation	Leukoplakia
Diffuse erythematous type	Oral lichen planus
Granular type (inflammatory papillary hyperplasia) *	Lupus erythematosus
Median rhomboid glossitis (MRG)	**Chronic mucocutaneous candidosis (CMC)**
Angular cheilitis	Congenital/ Familial CMC, diffuse type
Cheilocandidosis	Acquired CMC, focal, or diffuse type
Juxtavermillion candidosis	Endocrinopathy associated
**Oropharyngeal candidosis**	Hypothyroidism, hypoparathyroidism,
Dangerous sign of immunosuppression	MG-thymoma, Addison’s disease, etc.
**Suppurative oral candidosis**	Immunosuppression associated
Focal necrotizing ulcerative mucositis or osteomyelitis	Diabetes, iron deficiency, HIV/AIDS, neutropenia, etc.

MG, myasthenia gravis. * The granular type (inflammatory papillary hyperplasia) candidosis affects not only denture wearers but also immunodeficient patients without dentures.

## Data Availability

This review did not report any unpublished studies.

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
