# Peer review of "Oral Candidosis: Pathophysiology and Best Practice for Diagnosis, Classification, and Successful Management"

_jof, 2021, doi:10.3390/jof7070555_

Round 1
Reviewer 1 Report
This manuscript describes the pathophysiology of oral candidiasis, including the authors' research, and provides recommendations for the diagnosis, practical classification, and successful management of patients with this disease.
This is a comprehensive and systematic manuscript on the clinical aspects of oral candidiasis.
I believe that it is acceptable to publish the paper in its current form in the Journal of Fungi.
Author Response
My sincere thanks for your kind comments.
Reviewer 2 Report
The manuscript is a very well organized, extensive and detailed review of all relevant aspects of oral candidosis, from a clinical prespective. As a biologist focused on recurrence of infectious diseases, this reviewer greatly appreciated the effort since it summarizes the most important features to understand the pathophysiology of the disease, as well as treatment strategies - and why these many times are not successful. Tables are well constructed and the final message is helpful and open venues for new studies.
I have only a minor suggestion:
lines 58-59, line 116, line 346: italicize species name
Author Response
All species names in the manuscript have been italicized.
Reviewer 3 Report
This manuscript is a review of oral candidosis, its pathophysiology, diagnosis, classification and management. There have been many reviews of candidosis over the years and so any new contribution in this high impact journal must be well-reasoned, accurate, supported by primary literature and add a new perspective. This review meets some, but not all, of these criteria. The manuscript appears to be one clinician’s view of candidosis based primarily on their clinical experience. It does add a new perspective – focussing to a large degree on iron deficiency as a factor predisposing to oral candidosis, and CMC. However, the review does not meet some criteria in that in places it lacks accuracy, and is not supported by primary literature.
Major comments
- Much of the review cites other reviews (e.g. references [4,5,24) or text books (e.g. references [2,65]) to support its statements. This is not appropriate for a high-quality review – the author should base statements on a critical examination of the primary research literature.
- A lot of the literature cited is very old from the ‘70s, ‘80s and ‘90s. While it is always good to go back to seminal articles, they must be considered in their original context – the state of science at the time. Techniques and methodologies change and improve and so recent literature must also be considered.
- Some statements, especially in the introduction, are unsupported by references and so it is difficult to guage their accuracy (some examples are given below)
- Much of the discussion of clinical conditions and their causes, revolves around case reports. In order to ascribe underlying causes, larger studies of patients are needed.
- There is considerable reference to CMC. This discussion should include recent studies on the role of IL-17 IL-22 and STAT1 deficiencies in CMC.
- The section on Candida-associated denture stomatitis (lines 401-410) would benefit from clinical images, as for the other presentations.
- Lines 431-442, oropharyngeal candidosis is not a systemic infection. There should be a clearer definition of oropharyngeal candidosis and separate section on systemic infection.
- I question whether “Mechanical means to remove heavy candida plaques from oral lesions” (lines 512-513, and 610-611) should be recommended.
- I think that more than one clinician should be involved in re-classifying oral candidosis conditions.
- Lines 608-609, what is the evidence to propose limiting a carbohydrate-rich diet or sugar-rich drinks in order to prevent oral candidosis?
Minor comments
- Line 22, I don’t think that treatment failure or recurrence is inevitable.
- Line 24, replace “through” with ‘to’.
- Line 32, replace “present both as” with ‘be both’.
- Line 34, where is the evidence that most Candida “species cannot live at the temperature of the human body”?
- Line 36, where is the evidence that “Candida species live harmlessly in the eyes”?
- Lines 37-41 give the prevalence of albicans in the oral cavities of various population groups, but the references cited are old reviews. Are these numbers based on up-to-date studies?
- Lines 47-55 discuss the increased incidence of infections caused by non-albicans Candida species, yet the references cited are from 2002, 1990, 2006, 2016, 1984, 2014, 2008, 2010, 2001 and 2007 (with the 2016 article itself being a review). What has happened in the last five years?
- Lines 58-59, the Candida species should be in italics.
- Line 68, why refer to lysozyme which targets the peptidoglycan on bacterial cell walls?
- Line 91, replace “falls pit” with ‘pit falls’.
- Line 104, replace “practice” with ‘practise’.
- Lines 117-123 contrast early and more recent theories on the etiology of oral candidosis, yet the references cited ([2-4, 11-14] and [2,4, 12-14]) are the same, why?
- Line 126, where is the evidence that lysozyme, which cleaves the bond between N-acetylglucosamine and N-acetylmuramic acid in peptidoglycan, is antifungal?
- Line 140, why do psychotropic drugs predispose to oral Candida colonization and candidosis?
- Line 148, replace “is so-called a perfect pathogen” with ‘is a so-called perfect pathogen’.
- Lines 149-150, the morphological transition from yeast to hyphal or pseudohyphal forms is not “rapid microevolution” it is a phenotypic switch.
- Table 1, lines 164 and 165 replace “PH” with ‘pH’.
- Line 151, “fuzzy coat” is not a scientific term ‘fibrillar layer’ is sufficient.
- Lines 154-156, the discussion of high frequency switching should refer specifically to the white and opaque forms and the more recent GUT phenotype.
- Lines 164-167, where is the primary evidence that albicans can grow at pH 2.0? It is unlikely to be in references 22-24.
- Lines 172-178, this section needs to cite references, especially for “sugar is therefore a strong inducer for Candida germ-tube formation”. By “sugar” does the author mean sucrose or glucose? It is my understanding that N-acetylglucosamine induces germ-tube formation.
- Line 179, what are “custom-made beverages”?
- Line 180, replace “not a few number” with ‘many’.
- Lines 18186, how does an increase in glucose concentration decrease the activity of antifungal agents? Is this in vitro or in vivo?
- Table 2, what is “persorption”?
- Line 199, “Higgs and Wells” are mentioned, but no reference to their study is given.
- Line 205, it would be good to give clinical definitions of iron deficiency, anaemia and describe accepted treatment of iron deficency.
- Line 230, levels of what enzymes?
- Line 236, replace “could significantly implicate” with ‘could be implicated’.
- Line 239, please cite “my previous study”.
- Line 248, I think the study was by Lai et al, not just Lai and Lu.
- Lines 265-268, it is difficult to understand to what the various CD4 counts relate.
- Lines 307-308, how is an “active lesion” differentiated from an “old lesion” on clinical examination?
- Lines 312-316, provide more details of the tests and kits used to identify Candida
- Line 449, ‘cellulitis’ is spelled incorrectly, also cellulitis is a bacterial infection.
- Line 471, provide evidence that Candida infection is the leading cause of chronic oral ulcers in critically ill patients. I find this surprising.
- Line 474, I am also surprised that ulcers are covered by a “thick shiny organised biofilm”, I would expect ulcers to be covered by an inflamed infected epithelium.
- Line 481, replace “oropharygeal” with ‘oropharyngeal’.
- Lines 491-492, this sentence does not make sense.
- Line 564, insert ‘dose’ after “once daily”.
- The species names in the titles of the references should be in italics
- There is underlining in references 72 and 74.
- Why do references 75 and 76 have the same title?
Author Response
Response to Reviewer 3: I have tried my best to include a more comprehensive review of recent literature about oral candidosis.
- For major comments:
- 22 new references including recent literature from 2000-2020 have been included, and information about the role of IL-17, IL-22 and STAT1 deficiency in CMC has been added and discussed.
- Most statements, especially those in the introduction, are supported by references.
- The present researcher suggests that oropharyngeal candidosis and rare suppurative oral candidosis are an ominous sign of invasive candidosis and systemic candidemia, as exemplified in Figures 5 and 6. Oropharyngeal candidosis is not a systemic infection; however, it is a reliable marker for esophageal candidosis and candidemia in immunocompromised patients with fever, dysphagia, hypotension, leukocytosis, and refractory to previous broad-spectrum antibiotic therapy, endophthalmitis, suppurative phlebitis, or existed multifocal candidosis.
- In her over 30 years of clinical experiences, the present researcher often used a small spatula or a periosteal elevator to remove heavy candidal plaques from oral lesions, especially the tongue dorsum as the primary reservoir for oral Candida colonization and the initiating point of infection for the majority of the clinical forms of oral candidosis. The mechanical means can provide much help to promote antifungal action and speed healing, just as the procedure of debridement in the successful management of dentoalveolar abscess to reduce the bacterial infection, promote healing and improve the patient’s outcome.
- For minor comments:
Changes have been made according to your valuable suggestions. Further clarifications regarding some of your comments are provided as follows:
Answer to minor comment No. 1: Based on our previous research and over 30 years of clinical practice in a medical center, the treatment failure or recurrence of oral candidosis was not uncommon because of the inability to correct underlying risk factors, wrong diagnosis, inaccurate prescription of antifungal agents, or lack of patient education on using antifungals by many physicians and dentists. The statement was also supported by studies in the references 2, 19, 20, and 21.
Answer to minor comment No. 5: Candida albicans is also a commensal fungus of the normal conjunctival flora and an occasional contaminant of contact lenses and eye-banked donor corneas. C. albicans and other fungal eye infections are most commonly acquired following trauma or surgery, during hospitalization, and with immunosuppression. The normal flora of the eye plays an important role in maintaining ocular homeostasis by various mechanisms as oral cavity. Eyes are also protected by tears, which moisten them and clean out dirt, dust, and other irritants that get past the defenses of the eyelashes and eyelids. Tears also help protect against infection.
Answer to minor comment No. 11: The spelling of “practice” is remained, because the JoF guidelines for authors states that American English or UK English are fine so long as there is consistency.
Answer to minor comments No. 7 and 13: Saliva comprises many components of adaptive and innate immune response crucial for local host defenses. Saliva contains many nonspecific antimicrobial factors including antifungal actions such as lysozyme, lactoperoxidase, lactoferrin and histidine-rich polypeptides, which help to keep the oral Candida population under control (References 2, 5, 7). The elevated levels of IgA, secretory IgA (sIgA) and lysozyme demonstrated in patients with Candida-associated denture stomatitis may play a protective role (References 5, 26). Salivary IgA can also enhance the non-specific antimicrobial effect of lactoperoxidase system (Reference 26].
Answer to minor comment No.14: One of the important side effects from psychotropic drugs is xerostomia, which is a risk factor of oral Candida colonization and candidosis.
Answer to minor comments No. 20 and 21: Candida is aciduric/acidophilic, and can thrive in very low pH (2.0~4.0) media, which may promote its adhesion and colonization. Strong inducers of germ-tube formation for C. albicans include sugar, serum, ammonium ions or N-acetylglucosamine. Candida can grow well through a carbohydrate-rich diet or in sugar-supplemented mixed saliva. In vitro studies in Samaranayake et al. showed that using glucose-supplemented mixed saliva with added antibiotics to suppress bacterial growth, reported a rapid decline in pH from 7.5 to 3.2 over a 48-hour period accompanied by yeast growth (Reference 5). Clinically, we found that many young individuals developed angular cheilitis, atrophic glossitis, diffuse erythematous candidosis and also rampant cervical caries (“soda teeth”) that alerted the present researcher to the perception of diabetes mellitus and iron deficiency (ID) in these individuals (Figure 1). Carbohydrate includes sugar and starch. Sugar refers to monosaccharide (glucose and fructose) and disaccharide (sucrose). Appropriate references 2, 5, 42, and 43 have been cited.
Answer to minor comment No. 25: “Persorption” means the passage of yeasts from the intact mucosa into the blood stream. Studies reported that Candida could persorb through intact gastrointestinal mucosa and cause transient candidemia. It reflects the action of fungal hydrolytic enzymes; however, the phenomenon has been disputed (Reference 5). Therefore, “persorption” is deleted from Table 2 and a new “Table 2” has been re-organized.
Answer to minor comment No. 32: Clinical research evidence suggests that the location and intensity of candidosis in HIV patients are closely associated with a low CD4 count trend. When the CD4 count is progressively decreased, it will increase the risk of opportunistic candidal infections from oral and vaginal candidosis (CD4 count less than 400 cells/mm3), to oropharyngeal candidosis (CD4 count less than 200 cells/mm3), and then esophageal candidosis or systemic candidemia (CD4 count less than 30 cells/mm3) (Reference 67). This result is based on clinical observation, and the use of CD4 cell count as a marker of clinical disease progression is well established in HIV infection (References 11, 67).
Answer to minor comment No. 33: An “active lesion” means an active “fresh” lesion. For correct interpretation of a biopsy or smear, obtaining an appropriate biopsy or smear is important. It depends on (1) the selection of biopsy site (most severe and significant change in the lesion, such as fresh bullae, not an old ulcer lesion for oral pemphigus vulgaris, (2) the procedures used, and (3) the proper submission of the biopsy or smear.
Answer to minor comment No. 35: The “cellulitis” is usually caused by bacterial infection, but it can also be caused by fungal infection, such as orbital cellulitis by mucormycosis and aspergillosis, which can be the cause of life-threatening invasive orbital infections in immunocompromised patients. Recently, we've seen reports from India of infections with rhino-orbito-cerebral or nasopalatine mucormycosis, often termed "black fungus", in patients with Covid-19, or who are recovering from the coronavirus. Suppurative candidosis is a rare clinical entity that may show necrotizing ulcerative mucositis, submucosal abscess, cellulitis or even osteomyelitis of jawbones (References 96, 97).
Answer to minor comment No. 36 and major comment No. 8: Two new figures (8 and 9, highlighted in yellow in lines 574 and 612) and their legends are added to help with the reader’s understanding.
I hope the above responses have adequately addressed the reviewers’ comments to the manuscript. My sincere appreciation and I look forward to the publication of my revised manuscript in Journal of Fungi.
Thank you for your time.
Sincerely yours,
Shin-Yu Lu, Associate Professor
Director of Oral Medicine, Oral Pathology and Family Dentistry
Kaohsiung Chang Gung Memorial Hospital, Taiwan

Round 2
Reviewer 3 Report
This is a revised version of manuscript jof-122655 for which I was reviewer 3. The author has addressed some of my concerns, but some remain. Below, I refer to the comments I made for the original manuscript. I then have added some additional comments on the new text in the revised manuscript.
Major comments
- The addition of 22 references is welcomed. There are still some instances where there is no primary research cited. See major comment 10 and minor comments 5, 6, 13, 20 and 36.
- The addition of more recent references is welcomed.
- See my response under 1 above.
- The author has not responded to my major point 4: Much of the discussion of clinical conditions and their causes, revolves around case reports. In order to ascribe underlying causes, larger studies of patients are needed.
- There is now a good discussion of IL-17, IL-22, and STAT1 in CMC.
- The author has not responded to my major point 6: The section on Candida-associated denture stomatitis (lines 401-410) would benefit from clinical images, as for the other presentations.
- I do not think that oropharyngeal candidosis is a sign of systemic candidosis. Systemic candidosis leads to infections of organs such as the kidneys, it does not lead to infections of the oropharynx. Oropharyngeal candidosis may be a risk factor for systemic candidosis. The section on oropharyngeal candidosis still lacks a description of the presentation – lesions on the back third of the tongue, the soft palate, the side and back walls of the throat, and the tonsils. Indeed, the presentations in Figures 5 A, C, E, and G are not typical of oropharyngeal candidosis.
- Response to comment 8 should be verified by another clinician.
- The author has not responded to my major point 9: I think that more than one clinician should be involved in re-classifying oral candidosis conditions.
- The author has not responded to my major point 10: Lines 608-609, what is the evidence to propose limiting a carbohydrate-rich diet or sugar-rich drinks in order to prevent oral candidosis?
Minor comments
- I agree that the treatment failure or recurrence of oral candidosis is not uncommon, what I object to is the use of the word “inevitable” in line 22 of the Abstract. This means that there will always be treatment failure or recurrence. This is not the case – for some patients, treatment works and there is no recurrence. Therefore, in line 22, “inevitable” should be replaced with ‘common’.
- OK
- Line 32, delete “as”.
- OK
- I am aware that fungi can cause eye infections, I want the author to cite a paper that reports Candida species to be normal commensals (i.e. with no sign of infection) of the eye.
- The three references cited for prevalence of Candida albicans in different populations are reviews and not primary research papers.
- OK, some more recent papers have been added.
- OK
- The sentence on lines 68-72 “The three major species …..acrylic surfaces [5-7].” is explaining why albicans, C. tropicalis and C. glabrata cause oral candidosis more often than other Candida species. So why can these species evade histidine-rich peptides and lysozyme more effectively than other Candida species?
- OK
- OK
- OK
- I am aware of the antimicrobial factors in saliva. I would like the authors to cite a primary research publication that reports that lysozyme has antifungal activity.
- I think that dry mouth is more frequently a side effect of antipsychotic and antidepressant medication rather than psychotropic drugs in general.
- OK
- OK
- OK
- OK
- OK
- I am well aware that albicans is acidogenic and aciduric. The fact that it can survive at low pH does not mean that it grows well at pH 2.0. The author should cite the primary research publication (not a text book) that reports that C. albicans can grow well (not “well grow”) at pH 2.0.
- In the revised manuscript, the sentence on line 210, “Sugar is therefore a strong inducer for Candida germ-tube formation [2,5,42,43]” does not logically follow from the preceding sentences and I suggest deleting it. Indeed, there are research papers that show that glucose promotes yeast growth and not germ-tube formation.
- OK
- OK
- OK
- It is good that “persorption” has been deleted.
- OK
- There is now a good definition and description of iron deficiency.
- OK
- OK
- OK
- OK
- The author has explained what was meant, but has not altered the manuscript which is confusing. So, consider replacing: “When the CD4 count is progressively decreased, it will increase the risk of opportunistic candidal infections from oral and vaginal candidosis as CD4 count less than 400 cells/mm3 to oropharyngeal candidosis as CD4 less than 200 cells/mm3, and then esophageal candidosis or systemic candidemia as CD4 less than 30 cells/mm3 [67].” With ‘As the CD4 count progressively decreases, the risk of opportunistic candidal infections increases. When the CD4 count is less than 400 cells/mm3 there is an increased risk of oral and vaginal candidosis, below 200 cells/mm3 there is risk of oropharyngeal candidosis, and when the CD4 count is less than 30 cells/mm3 there is an increased risk of esophageal candidosis or systemic candidemia [67].’
- OK
- The API 20C and ID 32C kits are quite old, the newer technique of MALDI-TOF mass spectrometry should be mentioned.
- It is still my understanding that cellulitis is a bacterial infection. The authors should refer to ‘cellulitis-like dermis infections’ caused by fungi.
- The author has still not cited a reference to support the statement that “candidal infection is often the leading cause of chronic oral ulcers in critically ill patients”.
- Is the author sure that the greyish coloration of the mucosa (line 571) is a biofilm (on top of the mucosa) and not infected mucosa (like pseudomembraneous candidosis)?
- OK
- The author has not changed this sentence (lines 587-588). I think it will make sense if ‘to’ is added after “related”. Otherwise, the subject of the sentence is ID and acquired immunosuppression, which are not primary candida infections.
- OK
- OK
- OK
- OK
Additional new comments (due to inserted material)
- Lines 53-57, it is not clear whether the second half of this sentence is meant to explain the first half – why the shift in species is problematic for ICU patients. At present it does not because of the words “and furthermore”.
- Line 130, replace “comprises” with either ‘contains’ or ‘provides’.
- Lines 168-171, doesn’t “endocytosis” mean take up into cells rather than “penetrate between epithelial cells”.
- Line 202, why do SAPs (secreted aspartic proteinases) digest membranes which are made of phospholipids?
- Line 204, replace “mice model study newly discovered” with ‘mouse model study discovered’.
- Line 207, “In vitro” should be in italics.
- Line 209, replace “reported” with ‘resulted in’.
- Line 217, “In vivo” should be in italics.
- Line 218, insert ‘combined’ after “agents”.
- Lines 227-229 should define ID according to the WHO criteria not state what they were in study [19].
- Line 336, insert ‘producing’ after “(IL)-17”.
- Line 474, insert ‘is’ before “not”.
- Lines 474-475, delete “; moreover this is unlikely”.
Author Response
Response to Reviewer 3 (2nd round)
Thank you for your valuable comments. I have revised the manuscript according to Reviewer 3’s suggestions. All changes made are highlighted in the manuscript, and detailed responses are outlined as follows:
Response to major comments
- Addition of 16 new references, including 14 primary research papers. No. 8 and 9 address minor comment 5 that Candida species live harmlessly in the eyes; No. 10-13 for minor comment 6 addressing the prevalence of albicans in the oral cavities of various populations of people; No.22-24 address minor comment 13 about lysozyme with antifungal activity; No.54 and 55 address minor comment 20 that C. albicans and a few other species can grow well at a pH of less than 2.0. For minor comment 36, that “candidal infection is often the leading cause of chronic oral ulcers in critically ill patients” has been replaced with “candidal infection is often the leading cause of chronic oral ulcers after excluding other possible causes such as trauma, mucocutaneous diseases, syphilis, neoplasm, nutritional deficiency or drug reaction in critically ill patients.” There is no applicable reference found and added. The observation is “based on the present researcher’s clinical experience in a medical center over three decades.” This sentence has therefore been added on lines 595-598 to address this.
- A more recent reference and primary research have been added.
- See my response in comment 1 above.
- See my response in comment 1 above: 14 new primary research references with larger scale studies have been added. However, for cheilocandidosis, juxtavermillion candidosis, mucocutaneous candidosis, median rhomboid glossitis, hyperplastic candidosis, and suppurative candidosis, there have been no larger scale studies in this area due to their rarity.
- Thank you. (a good discussion of IL-17, IL-22, and STAT1 in CMC.)
- A new figure 4 and its legend about Candida-associated denture stomatitis have been added.
- (1)A description of the presentation about oropharyngeal candidosis, which may show lesions involving the back third of the tongue, the soft palate, the side and back walls of the throat, and the tonsils, has been added on lines 548-549. (2) To avoid misunderstanding, the author has replaced “It is a dangerous sign that may remind clinicians of potentially systemic candidosis,” with “Oropharyngeal candidosis may be a risk factor for systemic candidosis,” on lines 554-555. (3) I would like to explain that the presentations in Figure 5 were a clinically typical oropharyngeal candidosis with high fever and potential candidemia in a 24-year-old woman with systemic lupus erythematosus and uremia. Her manifestations of oropharyngeal candidosis involving pan-oral mucosa, the back third of the tongue, the soft palate, the side and back walls of the throat, and the tonsils can be clearly seen. However, her sore stomatitis and mouth-opening limitation made it very difficult to take clear photos over the side and back walls of the throat and the tonsil.
- The author has addressed this in the previous revision. I have successfully applied this mechanical means to remove heavy candidal plaque from oral lesion for decades and it works well in helping elimination of the number of pathogens and speeding healing (shown in Figure 10), as the maintenance of good oral and denture hygiene is crucial in management of oral candidosis and successful management of suppurative candidosis by debridement (shown in Figure 7) may enhance the efficacy of antifungal agents and create an environment to promote oral epithelial healing. After a comprehensive review of recent literature about oral candidosis, no other academic clinicians also perform an unverified clinical approach. It may be due to its rarity or ignorance in using oral pathology and oral medicine to improve patient’s care in their daily practice.
- I appreciate the reviewer’s focus on this issue. As indicated prior, my approach as a single-authored academic clinician is not without precedence (cf. Lehner, 1966; Odds, 1988; Lynch, 1994). Adding another clinician at this stage is impossible, particularly since I am recognized as one of the primary international experts in this specific area. However, I have made a minor change in this latest revision indicating this as a call for future research (on lines 634-637): “However, a limitation of this reclassification is that it is based on single clinician’s observations and experience over several decades of practice. Future research in this area would benefit from more clinicians’ opinions to keep this reclassification updated.”
- For a clearer explanation, the author has replaced “Liming a daily carbohydrate-rich diet and sugar-rich drinks is helpful to heal and prevent oral candidosis” with “Glucose promotes yeast growth and a high carbohydrate diet enhances its adherence to oral epithelial cells. Limiting their consumption is helpful in the control of oral Candida colonization and infection” on lines 755-758. In addition, the reasons for the above statement have been discussed on lines 205-235 and supported by primary research [reference 51,52,56] for the issue “Dietary carbohydrates can modulate albicans biofilm development by affecting its virulence factors and structural features [reference 51]”.
Response to minor comments
- Line 22: Corrected.
- OK
- Line 32: corrected
- OK
- New references No. 8 and 9 for “Candida species live harmlessly in the eyes” have been added.
- Four new primary research papers No. 10-13 for prevalence of albicans in the oral cavities of different populations are added.
- OK
- OK
- The reasons “why the three species of albicans, C. tropicalis and C. glabrata can evade lysozyme more effectively than other Candida species?” are discussed on lines 75-80 and three new references No. 22-24 have also been added: “Tobgi and Samaranayake and MacFarlane reported a significant dose-response relationship between lysozyme concentration and fungicidal activity. When different species of Candida were tested, C. krusei and C. parapsilosis were found to be most sensitive to lysozyme, compared with C. albicans and C. glabrata least sensitive [22]. The relatively high resistance of C. albicans and C. glabrata to lysozyme may partly explain their high oral prevalence both among carriers and in patients with oral candidosis [22-24].”
- OK
- OK
- OK
- Three primary research papers (No.22-24) addressing lysozyme with antifungal activity have been added.
- The “psychotropic drugs” have been replaced with “antipsychotic and antidepressant medications” on lines 162-163.
- OK
- OK
- OK
- OK
- OK
- A primary research paper (reference 54) addressing “C. albicans and a few other species can grow well at a pH of less than 2.0” has been added and discussed on lines 198-205.
- The original sentence “Sugar is therefore a strong inducer for Candida germ-tube formation” has been deleted and replaced with “Sugar may promote yeast growth [5,51,52,56,57]” to logically follow from the preceding sentences on lines 220-225 as “In vitro studies by Samaranayake et al. showed that using glucose-supplemented mixed saliva with added antibiotics to suppress bacterial growth, resulted in a rapid decline in pH from 7.5 to 3.2 over a 48-hour period accompanied by yeast growth [5]. These results suggest that sugar may promote yeast growth [5,51,52,56,57]”.
- OK
- OK
- OK
- OK (“persorption” has been deleted)
- OK
- Thank you. (A good definition of iron deficiency and description of iron deficiency)
- OK
- OK
- OK
- OK
- Corrected as reviewer 3 suggested on lines 322-328: “As the CD4 count progressively decreases, the risk of opportunistic candidal infections increases. When the CD4 count is less than 400 cells/mm3 there is an increased risk of oral and vaginal candidosis, below 200 cells/mm3 there is risk of oropharyngeal candidosis, and when the CD4 count is less than 30 cells/mm3 there is an increased risk of esophageal candidosis or systemic candidemia [80].”
- OK
- The newer technique of MALDI-TOF mass spectrometry and its references 90 and 91 have been added on lines 415-420.
- The author has made a change of “cellulitis” to “cellulitis-like dermis infections” caused by fungi on line 571.
- “Candidal infection is often the leading cause of chronic oral ulcers in critically ill patients.” has been replaced with “Candidal infection is often the leading cause of chronic oral ulcers after excluding other possible causes such as trauma, mucocutaneous diseases, syphilis, neoplasm, nutritional deficiency or drug reaction in critically ill patients.” This was based on the present researcher’s clinical experience in a medical center over three decades. There is no applicable reference found and added. Therefore, “based on the present researcher’s clinical experience in a medical center over three decades” has been added on lines 595-598 as clarification for readers.
- I appreciate the reviewer’s question whether the grayish coloration of the oral ulcers is a biofilm and not infected mucosa (like pseudomembranous candidosis). My explanation is that pseudomembranous candidosis usually shows lesions as multiple adherent white spots, which can join together to form a larger plaque like curdled milk or cottage cheese. Then the infected mucosa may appear inflamed, erythematous, bleeding or painful ulcers when plaques are wiped off. All the above features of pseudomembranous candidosis (as shown in Figure 10) are quite different from the presentations in Figure 9, which show the change of chronic oral ulcers covered by a thick shiny organized plaque with grayish color over lower labial mucosa, anterior gingiva, and tongue tip over 1 year in a critically ill 68-year-old woman with osteoarthritis and Reiter’s syndrome on many medications including prednisolone. Moreover, the thick shiny organized plaque with grayish color change is difficult to be wiped off. From observation of the progressive resolution of the lesion after antifungal therapy with fluconazole and nystatin tablets in Figures 9 A, B, C and D, the author would like to make sure that it was an organized biofilm-like material on the infected mucosa and not common pseudomembranous candidosis. The author therefore made a minor change in the description shown on lines 601-602 as “a large ulcer covered by a thick shiny plaque with grayish or yellowish color change as an organized biofilm-like material on the infected mucosa (Figure 9).”
- OK
- (The “to” is added after “related”)
- OK
- OK
- OK
- OK
Response to additional new comments:
- For a clearer explanation, the author has made a minor change with the inserted material on lines 54-60: “The shift in species with a higher incidence of candidosis caused by NAC species has emerged as a major bearing on the morbidity and management of patients who are cared for in intensive care units (ICUs) or have important risk factors such as hematologic malignancies, transplants, major abdominal surgery, and/or prolonged treatment with corticosteroid, as the increasing antifungal resistance along with a wide range of minimum inhibitory concentrations (MICs) to fluconazole between Candida species exists [16-21]. Therefore, the original ”the shift in species is problematic for ICU patients” has been deleted, and a new reference (No. 17, 2014) has been added to replace the old reference (No.11, 2008).
- Corrected on line 139.
- Corrected on lines 179-180.
- A clearer explanation has been discussed on lines 214-215: “SAPs (secreted aspartyl proteinases) and LPs (phospholipases) can digest and destroy cell membranes which are made of phospholipids with embedded proteins.” Therefore, “In addition to digesting and destroying cell membranes” has been replaced with “SAPs and LPs can digest and destroy cell membranes which are made of phospholipids with embedded protein.”
- Corrected on lines 218.
- Corrected on line 221.
- Corrected on line 222.
- Corrected on line 232.
- Corrected on line 233.
- Reference 58 about iron deficiency according to the WHO criteria is added on lines 242-248.
- Corrected on line 352.
- Corrected on line 496.
- Corrected on lines 496-457.
I hope the above responses have adequately addressed the reviewers’ comments to the manuscript. My sincere appreciation to your valuable and constructive feedback and I look forward to the publication of my revised manuscript in Journal of Fungi.
Thank you for your time.
Sincerely yours,
Shin-Yu Lu, Associate Professor
Director of Oral Medicine, Oral Pathology and Family Dentistry
Kaohsiung Chang Gung Memorial Hospital, Taiwan
